# The Incredible Shrinking Neural Network: New Perspectives on Learning Representations Through The Lens of Pruning

**Nikolas Wolfe, Aditya Sharma & Bhiksha Raj**
School of Computer Science
Carnegie Mellon University
Pittsburgh, PA 15213, USA
{nwolfe, bhiksha}@cs.cmu.edu, adityasharma@cmu.edu

**Lukas Drude**
Universitat Paderborn
drude@nt.upb.de

## Abstract

How much can pruning algorithms teach us about the fundamentals of learning representations in neural networks? A lot, it turns out. Neural network model compression has become a topic of great interest in recent years, and many different techniques have been proposed to address this problem. In general, this is motivated by the idea that smaller models typically lead to better generalization. At the same time, the decision of what to prune and when to prune necessarily forces us to confront our assumptions about how neural networks actually learn to represent patterns in data. In this work we set out to test several long-held hypotheses about neural network learning representations and numerical approaches to pruning. To accomplish this we first reviewed the historical literature and derived a novel algorithm to prune whole neurons (as opposed to the traditional method of pruning weights) from optimally trained networks using a second-order Taylor method. We then set about testing the performance of our algorithm and analyzing the quality of the decisions it made. As a baseline for comparison we used a first-order Taylor method based on the Skeletonization algorithm and an exhaustive brute-force serial pruning algorithm. Our proposed algorithm worked well compared to a first-order method, but not nearly as well as the brute-force method. Our error analysis led us to question the validity of many widely-held assumptions behind pruning algorithms in general and the trade-offs we often make in the interest of reducing computational complexity. We discovered that there is a straightforward way, however expensive, to serially prune 40-70% of the neurons in a trained network with minimal effect on the learning representation and without any re-training.

## 1 Introduction

In this work we propose and evaluate a novel algorithm for pruning whole neurons from a trained neural network without any re-training and examine its performance compared to two simpler methods. We then analyze the kinds of errors made by our algorithm and use this as a stepping off point to launch an investigation into the fundamental nature of learning representations in neural networks. Our results corroborate an insightful though largely forgotten observation by Mozer & Smolensky (1989a) concerning the nature of neural network learning. This observation is best summarized in a quotation from Segee & Carter (1991) on the notion of fault-tolerance in multilayer perceptron networks:

> Contrary to the belief widely held, multilayer networks are *not* inherently fault tolerant. In fact, the loss of a single weight is frequently sufficient to completely

> disrupt a learned function approximation. Furthermore, having a large number of weights *does not seem* to improve fault tolerance. [Emphasis added]

Essentially, Mozer & Smolensky (1989b) observed that during training neural networks do *not* distribute the learning representation evenly or equitably across hidden units. What actually happens is that a few, elite neurons learn an approximation of the input-output function, and the remaining units must learn a complex interdependence function which cancels out their respective influence on the network output. Furthermore, assuming enough units exist to learn the function in question, increasing the number of parameters does not increase the richness or robustness of the learned approximation, but rather simply increases the likelihood of overfitting and the number of noisy parameters to be canceled during training. This is evinced by the fact that in many cases, multiple neurons can be removed from a network with no re-training and with negligible impact on the quality of the output approximation. In other words, there are few bipartisan units in a trained network. A unit is typically either part of the (possibly overfit) input-output function approximation, or it is part of an elaborate noise cancellation task force. Assuming this is the case, most of the compute-time spent training a neural network is likely occupied by this arguably wasteful procedure of silencing superfluous parameters, and pruning can be viewed as a necessary procedure to "trim the fat."

We observed copious evidence of this phenomenon in our experiments, and this is the motivation behind our decision to evaluate the pruning algorithms in this study on the simple criteria of their ability to trim neurons *without* any re-training. If we were to employ re-training as part of our evaluation criteria, we would arguably *not* be evaluating the quality of our algorithm's pruning decisions per se but rather the ability of back-propagation trained networks to recover from faults caused by non-ideal pruning decisions, as suggested by the conclusions of Segee & Carter (1991) and Mozer & Smolensky (1989a). Moreover, as Fahlman & Lebiere (1989) discuss, due to the "herd effect" and "moving target" phenomena in back-propagation learning, the remaining units in a network will simply shift course to account for whatever error signal is re-introduced as a result of a bad pruning decision or network fault. So long as there are enough critical parameters to learn the function in question, a network can typically recover faults with additional training. This limits the conclusions we can draw about the quality of our pruning criteria when we employ re-training.

In terms of removing units without re-training, what we discovered is that predicting the behavior of a network when a unit is to be pruned is very difficult, and most of the approximation techniques put forth in existing pruning algorithms do not fare well at all when compared to a brute-force search. To begin our discussion of how we arrived at our algorithm and set up our experiments, we review of the existing literature.

## 2 LITERATURE REVIEW

Pruning algorithms, as comprehensively surveyed by Reed (1993), are a useful set of heuristics designed to identify and remove elements from a neural network which are either redundant or do not significantly contribute to the output of the network. This is motivated by the observed tendency of neural networks to overfit to the idiosyncrasies of their training data given too many trainable parameters or too few input patterns from which to generalize, as stated by Chauvin (1990).

Network architecture design and hyperparameter selection are inherently difficult tasks typically approached using a few well-known rules of thumb, e.g. various weight initialization procedures, choosing the width and number of layers, different activation functions, learning rates, momentum, etc. Some of this "black art" appears unavoidable. For problems which cannot be solved using linear threshold units alone, Baum & Haussler (1989) demonstrate that there is no way to precisely determine the appropriate size of a neural network a priori given any random set of training instances. Using too few neurons seems to inhibit learning, and so in practice it is common to attempt to over-parameterize networks initially using a large number of hidden units and weights, and then prune or compress them afterwards if necessary. Of course, as the old saying goes, there's more than one way to skin a neural network.

### 2.1 NON-PRUNING BASED GENERALIZATION & COMPRESSION TECHNIQUES

The generalization behavior of neural networks has been well studied, and apart from pruning algorithms many heuristics have been used to avoid overfitting, such as dropout (Srivastava et al.

(2014)), maxout (Goodfellow et al. (2013)), and cascade correlation (Fahlman & Lebiere (1989)), among others. Of course, while cascade correlation specifically tries to construct of minimal networks, many techniques to improve network generalization do not explicitly attempt to reduce the total number of parameters or the memory footprint of a trained network per se.

Model compression often has benefits with respect to generalization performance and the portability of neural networks to operate in memory-constrained or embedded environments. Without explicitly removing parameters from the network, weight quantization allows for a reduction in the number of bytes used to represent each weight parameter, as investigated by Balzer et al. (1991), Dundar & Rose (1994), and Hoehfeld & Fahlman (1992).

A recently proposed method for compressing recurrent neural networks (Prabhavalkar et al. (2016)) uses the singular values of a trained weight matrix as basis vectors from which to derive a compressed hidden layer. Øland & Raj (2015) successfully implemented network compression through weight quantization with an encoding step while others such as Han et al. (2016) have tried to expand on this by adding weight-pruning as a preceding step to quantization and encoding.

In summary, we can say that there are many different ways to improve network generalization by altering the training procedure, the objective error function, or by using compressed representations of the network parameters. But these are not, strictly speaking, examples of techniques to reduce the number of parameters in a network. For this we must employ some form of pruning criteria.

## 2.2 Pruning Techniques

If we wanted to continually shrink a neural network down to minimum size, the most straightforward brute-force way to do it is to individually switch each element off and measure the increase in total error on the training set. We then pick the element which has the least impact on the total error, and remove it. Rinse and repeat. This is extremely computationally expensive, given a reasonably large neural network and training set. Alternatively, we might accomplish this using any number of much faster off-the-shelf pruning algorithms, such as Skeletonization (Mozer & Smolensky (1989a)), Optimal Brain Damage (LeCun et al. (1989)), or later variants such as Optimal Brain Surgeon (Hassibi & Stork (1993)). In fact, we borrow much of our inspiration from these algorithms, with one major variation: Instead of pruning individual weights, we prune entire neurons, thereby eliminating all of their incoming and outgoing weight parameters in one go, resulting in more memory saved, faster.

The algorithm developed for this paper is targeted at reducing the total number of neurons in a trained network, which is one way of reducing its computational memory footprint. This is often a desirable criteria to minimize in the case of resource-constrained or embedded devices, and also allows us to probe the limitations of pruning down to the very last essential network elements. In terms of generalization as well, we can measure the error of the network on the test set as each element is sequentially removed from the network. With an oracle pruning algorithm, what we expect to observe is that the output of the network remains stable as the first few superfluous neurons are removed, and as we start to bite into the more crucial members of the function approximation, the error should start to rise dramatically. In this paper, the brute-force approach described at the beginning of this section serves as a proxy for an oracle pruning algorithm.

One reason to choose to rank and prune individual neurons as opposed to weights is that there are far fewer elements to consider. Furthermore, the removal of a single weight from a large network is a drop in the bucket in terms of reducing a network's core memory footprint. If we want to reduce the *size* of a network as efficiently as possible, we argue that pruning neurons instead of weights is more efficient computationally as well as practically in terms of quickly reaching a hypothetical target reduction in memory consumption. This approach also offers downstream applications a realistic expectation of the minimal increase in error resulting from the removal of a specified percentage of neurons. Such trade-offs are unavoidable, but performance impacts can be limited if a principled approach is used to find the best candidate neurons for removal.

It is well known that too many free parameters in a neural network can lead to overfitting. Regardless of the number of weights used in a given network, as Segee & Carter (1991) assert, the representation of a learned function approximation is almost never evenly distributed over the hidden units, and thus the removal of any single hidden unit at random can actually result in a network fault. Mozer & Smolensky (1989b) argue that only a subset of the hidden units in a neural network actually

latch on to the invariant or generalizing properties of the training inputs, and the rest learn to either mutually cancel each other's influence or begin overfitting to the noise in the data. We leverage this idea in the current work to rank all neurons in pre-trained networks based on their effective contributions to the overall performance. We then remove the unnecessary neurons to reduce the network's footprint. Through our experiments we not only concretely validate the theory put forth by Mozer & Smolensky (1989b) but we also successfully build on it to prune networks to 40 to 60 % of their original size without any major loss in performance.

## 3 PRUNING NEURONS TO SHRINK NEURAL NETWORKS

As discussed in Section 1 our aim is to leverage the highly non-uniform distribution of the learning representation in pre-trained neural networks to eliminate redundant neurons, without focusing on individual weight parameters. Taking this approach enables us to remove all the weights (incoming and outgoing) associated with a non-contributing neuron at once. We would like to note here that in an ideal scenario, based on the neuron interdependency theory put forward by Mozer & Smolensky (1989a), one would evaluate all possible combinations of neurons to remove (one at a time, two at a time, three at a time and so forth) to find the optimal subset of neurons to keep. This is computationally unacceptable, and so we will only focus on removing one neuron at a time and explore more "greedy" algorithms to do this in a more efficient manner.

The general approach taken to prune an optimally trained neural network here is to create a ranked list of all the neurons in the network based off of one of the 3 proposed ranking criteria: a brute force approximation, a linear approximation and a quadratic approximation of the neuron's impact on the output of the network. We then test the effects of removing neurons on the accuracy and error of the network. All the algorithms and methods presented here are easily parallelizable as well.

One last thing to note here before moving forward is that the methods discussed in this section involve some non-trivial derivations which are beyond the scope of this paper. We are more focused on analyzing the implications of these methods on our understanding of neural network learning representations. However, a complete step-by-step derivation and proof of all the results presented is provided in the Supplementary Material as an Appendix.

### 3.1 BRUTE FORCE REMOVAL APPROACH

This is perhaps the most naive yet the most accurate method for pruning the network. It is also the slowest and hence possibly unusable on large-scale neural networks with thousands of neurons. This method explicitly evaluates each neuron in the network. The idea is to manually check the effect of every single neuron on the output. This is done by running a forward propagation on the validation set $K$ times (where $K$ is the total number of neurons in the network), turning off exactly one neuron each time (keeping all other neurons active) and noting down the change in error. Turning a neuron off can be achieved by simply setting its output to 0. This results in all the outgoing weights from that neuron being turned off. This change in error is then used to generate the ranked list.

### 3.2 TAYLOR SERIES REPRESENTATION OF ERROR

Let us denote the total error from the optimally trained neural network for any given validation dataset by $E$. $E$ can be seen as a function of $O$, where $O$ is the output of any general neuron in the network. This error can be approximated at a particular neuron's output (say $O_k$) by using the 2nd order Taylor Series as,

$$\hat{E}(O) \approx E(O_k) + (O - O_k) \cdot \left.\frac{\partial E}{\partial O}\right|_{O_k} + 0.5 \cdot (O - O_k)^2 \cdot \left.\frac{\partial^2 E}{\partial O^2}\right|_{O_k}, \tag{1}$$

When a neuron is pruned, its output $O$ becomes 0.

Replacing $O$ by $O_k$ in equation 1 shows us that the error is approximated perfectly by equation 1 at $O_k$. So:

$$\Delta E_k = \hat{E}(0) - \hat{E}(O_k) = -O_k \cdot \left.\frac{\partial E}{\partial O}\right|_{O_k} + 0.5 \cdot O_k^2 \cdot \left.\frac{\partial^2 E}{\partial O^2}\right|_{O_k}, \qquad (2)$$

where $\Delta E_k$ is the change in the total error of the network when exactly one neuron ($k$) is turned off. Most of the terms in this equation are fairly easy to compute, as we have $O_k$ already from the activations of the hidden units and we already compute $\frac{\partial E}{\partial O}|_{O_k}$ for each training instance during backpropagation. The $\frac{\partial^2 E}{\partial O^2}|_{O_k}$ terms are a little more difficult to compute. This is derived in the appendix and summarized in the sections below.

### 3.2.1 LINEAR APPROXIMATION APPROACH

We can use equation 2 to get the linear error approximation of the change in error due to the $k$th neuron being turned off and represent it as $\Delta E_k^1$ as follows:

$$\Delta E_k^1 = -O_k \cdot \left.\frac{\partial E}{\partial O}\right|_{O_k} \qquad (3)$$

The derivative term above is the first-order gradient which represents the change in error with respect to the output a given neuron. This term can be collected during back-propagation. As we shall see further in this section, linear approximations are not reliable indicators of change in error but they provide us with an interesting basis for comparison with the other methods discussed in this paper.

### 3.2.2 QUADRATIC APPROXIMATION APPROACH

As above, we can use equation 2 to get the quadratic error approximation of the change in error due to the $k$th neuron being turned off and represent it as $\Delta E_k^2$ as follows:

$$\Delta E_k^2 = -O_k \cdot \left.\frac{\partial E}{\partial O}\right|_{O_k} + 0.5 \cdot O_k^2 \cdot \left.\frac{\partial^2 E}{\partial O^2}\right|_{O_k} \qquad (4)$$

The additional second-order gradient term appearing above represents the quadratic change in error with respect to the output of a given neuron. This term can be generated by performing back-propagation using second order derivatives. Collecting these quadratic gradients involves some non-trivial mathematics, the entire step-by-step derivation procedure of which is provided in the Supplementary Material as an Appendix.

### 3.3 PROPOSED PRUNING ALGORITHM

Figure 1 shows a random error function plotted against the output of any given neuron. Note that this figure is for illustration purposes only. The error function is minimized at a particular value of the neuron output as can be seen in the figure. The process of training a neural network is essentially the process of finding these minimizing output values for all the neurons in the network. Pruning this particular neuron (which translates to getting a zero output from it will result in a change in the total overall error. This change in error is represented by distance between the original minimum error (shown by the dashed line) and the top red arrow. This neuron is clearly a bad candidate for removal since removing it will result in a huge error increase.

The straight red line in the figure represents the first-order approximation of the error using Taylor Series as described before while the parabola represents a second-order approximation. It can be clearly seen that the second-order approximation is a much better estimate of the change in error.

One thing to note here is that it is possible in some cases that there is some thresholding required when trying to approximate the error using the 2nd order Taylor Series expansion. These cases might arise when the parabolic approximation undergoes a steep slope change. To take into account such cases, mean and median thresholding were employed, where any change above a certain threshold was assigned a mean or median value respectively.

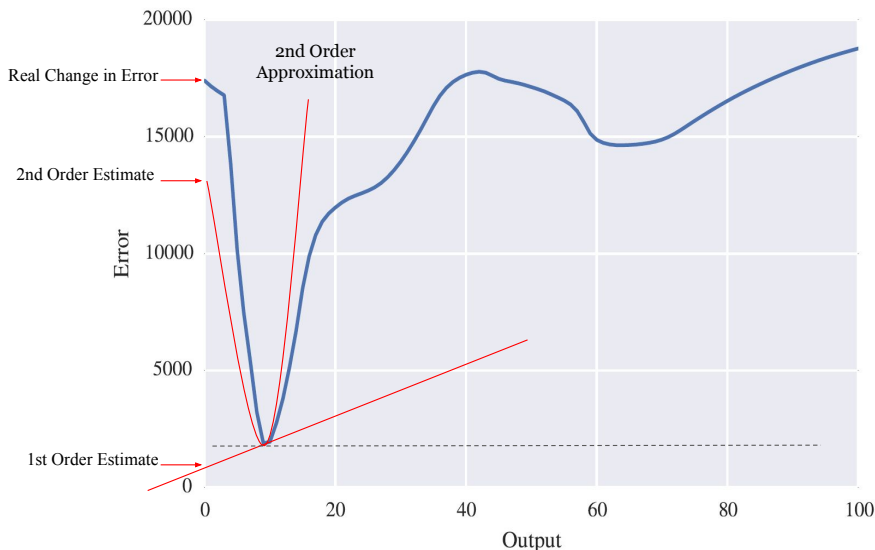

Figure 1: The intuition behind 1st & 2nd order neuron pruning decisions

Two pruning algorithms are proposed here. They are different in the way the neurons are ranked but both of them use $\Delta E_k$, the approximation of the change in error as the basis for the ranking. $\Delta E_k$ can be calculated using the Brute Force method, or one of the two Taylor Series approximations discussed previously.

The first step in both the algorithms is to decide a stopping criterion. This can vary depending on the application but some intuitive stopping criteria can be: maximum number of neurons to remove, percentage scaling needed, maximum allowable accuracy drop etc.

### 3.3.1 ALGORITHM I: SINGLE OVERALL RANKING

The complete algorithm is shown in Algorithm 1. The idea here is to generate a single ranked list based on the values of $\Delta E_k$. This involves a single pass of second-order back-propagation (without weight updates) to collect the gradients for each neuron. The neurons from this rank-list (with the lowest values of $\Delta E_k$) are then pruned according to the stopping criterion decided. We note here that this algorithm is intentionally naive and is used for comparison only.

**Data:** optimally trained network, training set
**Result:** A pruned network
initialize and define stopping criterion ;
perform forward propagation over the training set ;
perform second-order back-propagation without updating weights and collect linear and quadratic
 gradients ;
rank the remaining neurons based on $\Delta E_k$;
**while** *stopping criterion is not met* **do**
 | remove the last ranked neuron ;
**end**

**Algorithm 1:** Single Overall Ranking

### 3.3.2 ALGORITHM II: ITERATIVE RE-RANKING

In this greedy variation of the algorithm (Algorithm 2), after each neuron removal, the remaining network undergoes a single forward and backward pass of second-order back-propagation (without weight updates) and the rank list is formed again. Hence, each removal involves a new pass through

the network. This method is computationally more expensive but takes into account the dependencies the neurons might have on one another which would lead to a change in error contribution every time a dependent neuron is removed.

**Data:** optimally trained network, training set
**Result:** A pruned network
initialize and define stopping criterion ;
**while** *stopping criterion is not met* **do**

 perform forward propagation over the training set ;
 perform second-order back-propagation without updating weights and collect linear and
  quadratic gradients ;
 rank the remaining neurons based on $\Delta E_k$ ;
 remove the worst neuron based on the ranking ;

**end**

**Algorithm 2:** Iterative Re-Ranking

# 4 EXPERIMENTAL RESULTS

## 4.1 EXAMPLE REGRESSION PROBLEM

This problem serves as a quick example to demonstrate many of the phenomena described in previous sections. We trained two networks to learn the cosine function, with one input and one output. This is a task which requires no more than 11 sigmoid neurons to solve entirely, and in this case we don't care about overfitting because the cosine function has a precise definition. Furthermore, the cosine function is a good toy example because it is a smooth continuous function and, as demonstrated by Nielsen (2015), if we were to tinker directly with the weights and bias parameters of the network, we could allocate individual units within the network to be responsible for constrained ranges of inputs, similar to a basis spline function with many control points. This would distribute the learned function approximation evenly across all hidden units, and thus we have presented the network with a problem in which it could productively use as many hidden units as we give it. In this case, a pruning algorithm would observe a fairly consistent increase in error after the removal of each successive unit. In practice however, regardless of the number of experimental trials, this is not what happens. The network will always use 10-11 hidden units and leave the rest to cancel each other's influence.

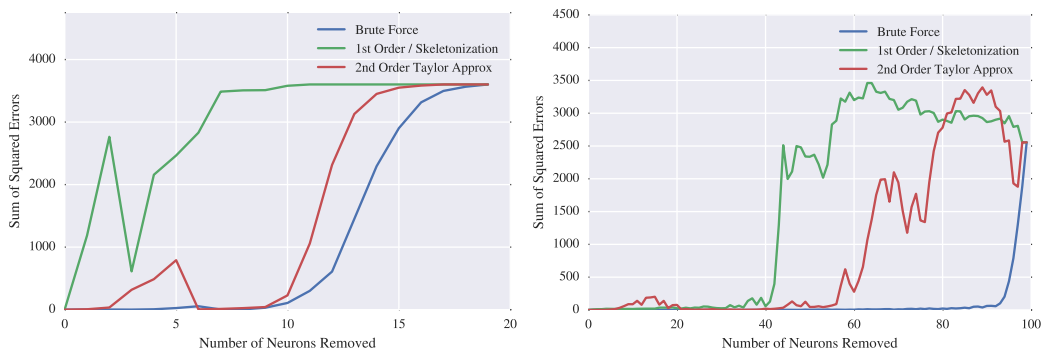

Figure 2: Degradation in squared error after pruning a two-layer network trained to compute the cosine function (**Left Network:** 2 layers, 10 neurons each, 1 output, logistic sigmoid activation, starting test accuracy: 0.9999993, **Right Network:** 2 layers, 50 neurons each, 1 output, logistic sigmoid activation, starting test accuracy: 0.9999996)

Figure 2 shows two graphs. Both graphs demonstrate the use of the iterative re-ranking algorithm and the comparative performance of the brute-force pruning method (in blue), the first order method (in green), and the second order method (in red). The graph on the left shows the performance of these algorithms starting from a network with two layers of 10 neurons (20 total), and the graph on the right shows a network with two layers of 50 neurons (100 total).

In the left graph, we see that the brute-force method shows a graceful degradation, and the error only begins to rise sharply after 50% of the total neurons have been removed. The error is basically constant up to that point. In the first and second order methods, we see evidence of poor decision making in the sense that both made mistakes early on, which disrupted the output function approximation. The first order method made a large error early on, though we see after a few more neurons were removed this error was corrected somewhat (though it only got worse from there). This is direct evidence of the lack of fault tolerance in a trained neural network. This phenomenon is even more starkly demonstrated in the second order method. After making a few poor neuron removal decisions in a row, the error signal rose sharply, and then went back to zero after the 6th neuron was removed. This is due to the fact that the neurons it chose to remove were trained to cancel each others' influence within a localized part of the network. After the entire group was eliminated, the approximation returned to normal. This can only happen if the output function approximation is not evenly distributed over the hidden units in a trained network.

This phenomenon is even more starkly demonstrated in the graph on the right. Here we see the first order method got "lucky" in the beginning and made decent decisions up to about the 40th removed neuron. The second order method had a small error in the beginning which it recovered from gracefully and proceeded to pass the 50 neuron point before finally beginning to unravel. The brute force method, in sharp contrast, shows little to no increase in error at all until 90% of the neurons in the network have been obliterated. Clearly first and second order methods have some value in that they do not make completely arbitrary choices, but the brute force method is far better at this task.

This also demonstrates the sharp dualism in neuron roles within a trained network. These networks were trained to near-perfect precision and each pruning method was applied *without* any re-training of any kind. Clearly, in the case of the brute force or oracle method, up to 90% of the network can be completely extirpated before the output approximation even begins to show any signs of degradation. This would be impossible if the learning representation were evenly or equitably distributed. Note, for example, that the degradation point in both cases is approximately the same. This example is not a real-world application of course, but it brings into very clear focus the kind of phenomena we will discuss in the following sections.

## 4.2 RESULTS ON MNIST DATASET

For all the results presented in this section, the MNIST database of Handwritten Digits by LeCun & Cortes (2010) was used. It is worth noting that due to the time taken by the brute force algorithm we rather used a 5000 image subset of the MNIST database in which we have normalized the pixel values between 0 and 1.0, and compressed the image sizes to 20x20 images rather than 28x28, so the starting test accuracy reported here appears higher than those reported by LeCun et al. We do not believe that this affects the interpretation of the presented results because the basic learning problem does not change with a larger dataset or input dimension.

## 4.3 PRUNING A 1-LAYER NETWORK

The network architecture in this case consisted of 1 layer, 100 neurons, 10 outputs, logistic sigmoid activations, and a starting test accuracy of 0.998.

### 4.3.1 SINGLE OVERALL RANKING ALGORITHM

We first present the results for a single-layer neural network in Figure 3, using the Single Overall algorithm (Algorithm 1) as proposed in Section 3. (We again note that this algorithm is intentionally naive and is used for comparison only. Its performance should be expected to be poor.) After training, each neuron is assigned its permanent ranking based on the three criteria discussed previously: A brute force "ground truth" ranking, and two approximations of this ranking using first and second order Taylor estimations of the change in network output error resulting from the removal of each neuron.

An interesting observation here is that with only a single layer, no criteria for ranking the neurons in the network (brute force or the two Taylor Series variants) using Algorithm 1 emerges superior, indicating that the 1st and 2nd order Taylor Series methods are actually reasonable approximations

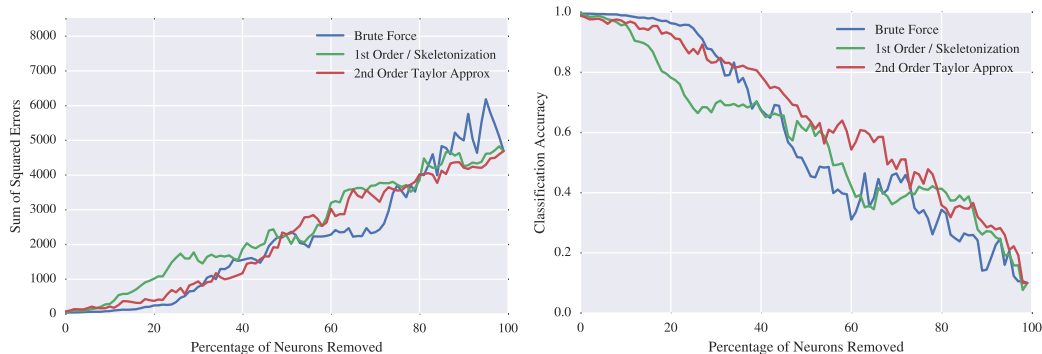

Figure 3: Degradation in squared error (left) and classification accuracy (right) after pruning a single-layer network using The Single Overall Ranking algorithm (**Network:** 1 layer, 100 neurons, 10 outputs, logistic sigmoid activation, starting test accuracy: 0.998)

of the brute force method under certain conditions. Of course, this method is still quite bad in terms of the rate of degradation of the classification accuracy and in practice we would likely follow Algorithm 2 which is takes into account Mozer & Smolensky (1989a)'s observations stated in the Related Work section. The purpose of the present investigation, however, is to demonstrate how much of a trained network can be theoretically removed without altering the network's learned parameters in any way.

### 4.3.2 ITERATIVE RE-RANKING ALGORITHM

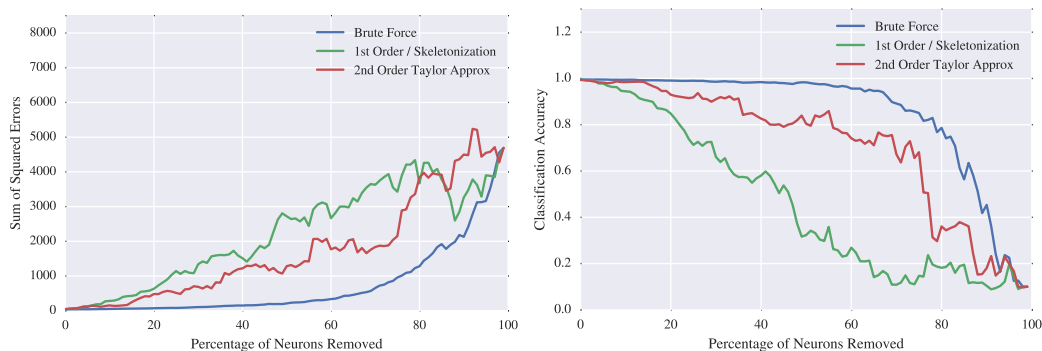

Figure 4: Degradation in squared error (left) and classification accuracy (right) after pruning a single-layer network the iterative re-ranking algorithm (**Network:** 1 layer, 100 neurons, 10 outputs, logistic sigmoid activation, starting test accuracy: 0.998)

In Figure 4 we present our results using Algorithm 2 (The iterative re-ranking Algorithm) in which all remaining neurons are re-ranked after each successive neuron is switched off. We compute the same brute force rankings and Taylor series approximations of error deltas over the remaining active neurons in the network after each pruning decision. This is intended to account for the effects of cancelling interactions between neurons.

There are 2 key observations here. Using the brute force ranking criteria, almost 60% of the neurons in the network can be pruned away without any major loss in performance. The other noteworthy observation here is that the 2nd order Taylor Series approximation of the error performs consistently better than its 1st order version, in most situations, though Figure 21 is a poignant counter-example.

### 4.3.3 VISUALIZATION OF ERROR SURFACE & PRUNING DECISIONS

As explained in Section 3, these graphs are a visualization of the error surface of the network output with respect to the neurons chosen for removal using each of the 3 ranking criteria, represented in

intervals of 10 neurons. In each graph, the error surface of the network output is displayed in log space (left) and in real space (right) with respect to each candidate neuron chosen for removal. We create these plots during the pruning exercise by picking a neuron to switch off, and then multiplying its output by a scalar gain value $\alpha$ which is adjusted from 0.0 to 10.0 with a step size of 0.001. When the value of $\alpha$ is 1.0, this represents the unperturbed neuron output learned during training. Between 0.0 and 1.0, we are graphing the literal effect of turning the neuron off ($\alpha = 0$), and when $\alpha > 1.0$ we are simulating a boosting of the neuron's influence in the network, i.e. inflating the value of its outgoing weight parameters.

We graph the effect of boosting the neuron's output to demonstrate that for certain neurons in the network, even doubling, tripling, or quadrupling the scalar output of the neuron has no effect on the overall error of the network, indicating the remarkable degree to which the network has learned to ignore the value of certain parameters. In other cases, we can get a sense of the sensitivity of the network's output to the value of a given neuron when the curve rises steeply after the red 1.0 line. This indicates that the learned value of the parameters emanating from a given neuron are relatively important, and this is why we should ideally see sharper upticks in the curves for the later-removed neurons in the network, that is, when the neurons crucial to the learning representation start to be picked off. Some very interesting observations can be made in each of these graphs.

Remember that lower is better in terms of the height of the curve and minimal (or negative) horizontal change between the vertical red line at 1.0 (neuron *on*, $\alpha = 1.0$) and 0.0 (neuron *off*, $\alpha = 0.0$) is indicative of a good candidate neuron to prune, i.e. there will be minimal effect on the network output when the neuron is removed.

### 4.3.4 VISUALIZATION OF BRUTE FORCE PRUNING DECISIONS

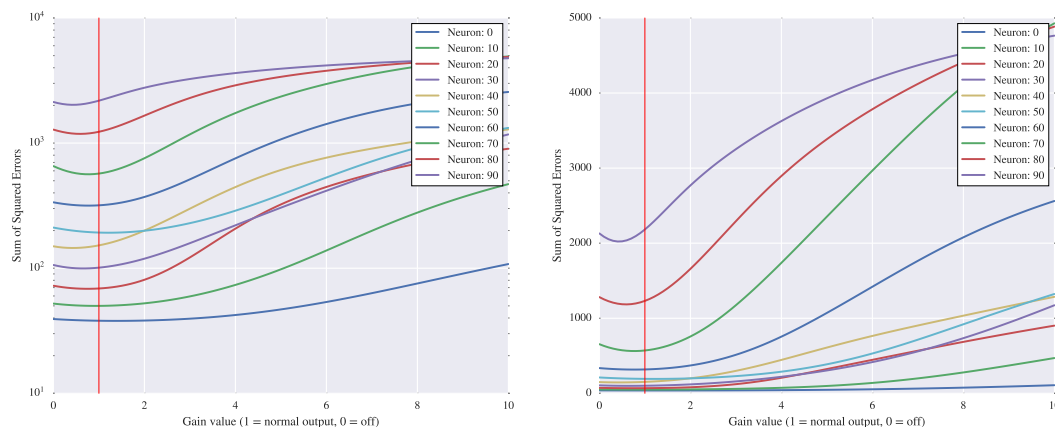

Figure 5: Error surface of the network output in log space (left) and real space (right) with respect to each candidate neuron chosen for removal using the brute force criterion; (**Network:** 1 layer, 100 neurons, 10 outputs, logistic sigmoid activation, starting test accuracy: 0.998)

In Figure **??**, we notice how low to the floor and flat most of the curves are. It's only until the 90th removed neuron that we see a higher curve with a more convex shape (clearly a more sensitive, influential piece of the network).

### 4.3.5 VISUALIZATION OF 1ST ORDER APPROXIMATION PRUNING DECISIONS

It can be seen in Figure 6 that most choices seem to have flat or negatively sloped curves, indicating that the first order approximation seems to be pretty good, but examining the brute force choices shows they could be better.

### 4.3.6 VISUALIZATION OF 2ND ORDER APPROXIMATION PRUNING DECISIONS

The method in Figure 7 looks similar to the brute force method choices, though clearly not as good (they're more spread out). Notice the difference in convexity between the 2nd and 1st order method

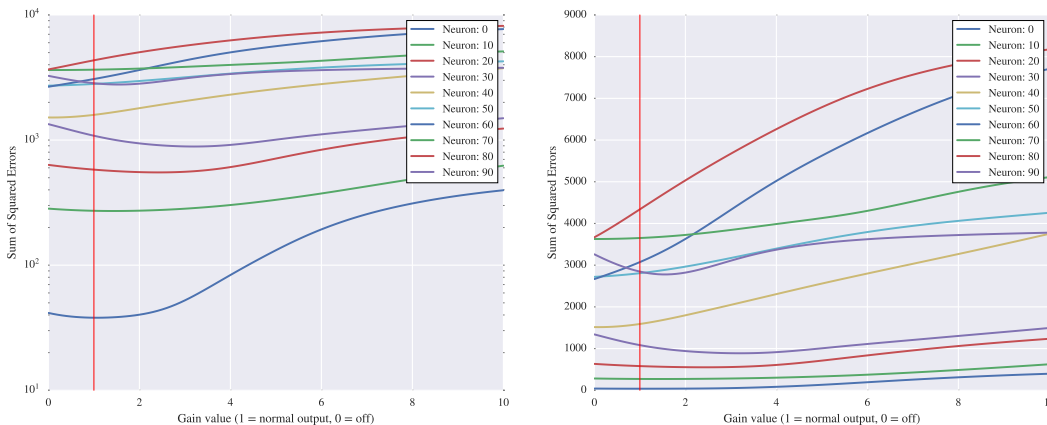

Figure 6: Error surface of the network output in log space (left) and real space (right) with respect to each candidate neuron chosen for removal using the 1st order Taylor Series error approximation criterion; (**Network:** 1 layer, 100 neurons, 10 outputs, logistic sigmoid activation, starting test accuracy: 0.998)

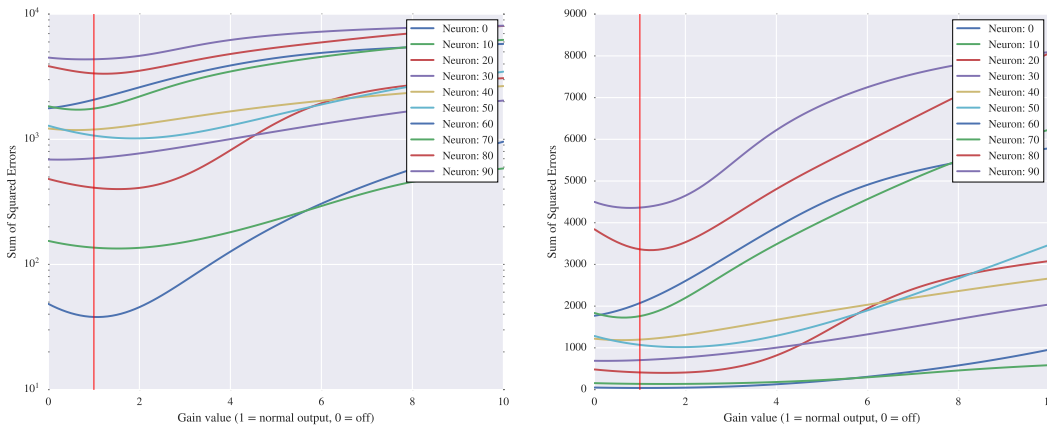

Figure 7: Error surface of the network output in log space (left) and real space (right) with respect to each candidate neuron chosen for removal using the 2nd order Taylor Series error approximation criterion; (**Network:** 1 layer, 100 neurons, 10 outputs, logistic sigmoid activation, starting test accuracy: 0.998)

choices. It's clear that the first order method is fitting a line and the 2nd order method is fitting a parabola in their approximation.

## 4.4 PRUNING A 2-LAYER NETWORK

The network architecture in this case consisted of 2 layers, 50 neurons per layer, 10 outputs, logistic sigmoid activations, and a starting test accuracy of 1.000.

### 4.4.1 SINGLE OVERALL RANKING ALGORITHM

Figure 8 shows the pruning results for Algorithm 1 on a 2-layer network. The ranking procedure is identical to the one used to generate Figure 3. (We again note that this algorithm is intentionally naive and is used for comparison only. Its performance should be expected to be poor.)

Unsurprisingly, a 2-layer network is harder to prune because a single overall ranking will never capture the interdependencies between neurons in different layers. It makes sense that this is worse

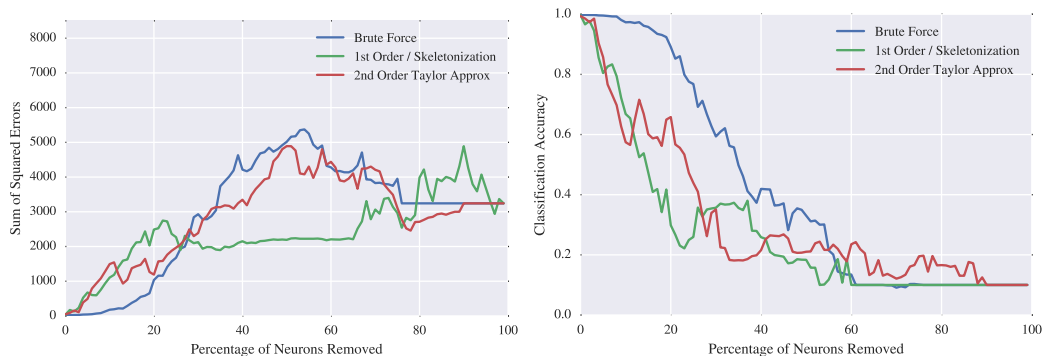

Figure 8: Degradation in squared error (left) and classification accuracy (right) after pruning a 2-layer network using the Single Overall Ranking algorithm; (**Network:** 2 layers, 50 neurons/layer, 10 outputs, logistic sigmoid activation, starting test accuracy: 1.000)

than the performance on the 1-layer network, even if this method is already known to be bad, and we'd likely never use it in practice.

### 4.4.2 ITERATIVE RE-RANKING ALGORITHM

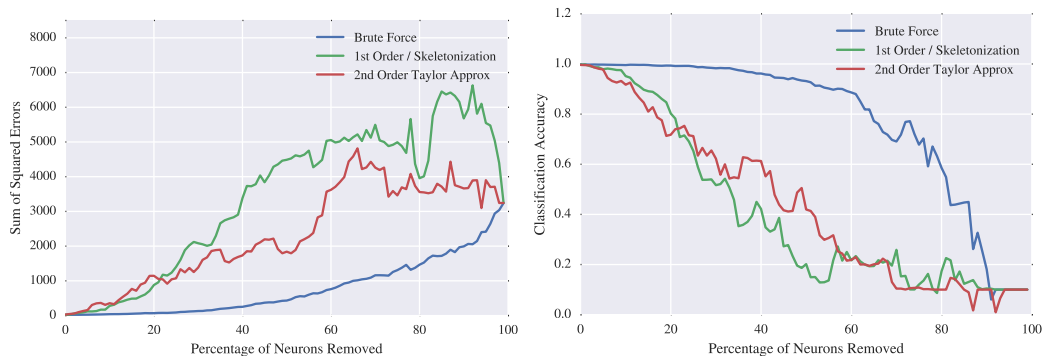

Figure 9: Degradation in squared error (left) and classification accuracy (right) after pruning a 2-layer network using the iterative re-ranking algorithm; (**Network:** 2 layers, 50 neurons/layer, 10 outputs, logistic sigmoid activation, starting test accuracy: 1.000)

Figure 9 shows the results from using Algorithm 2 on a 2-layer network. We compute the same brute force rankings and Taylor series approximations of error deltas over the remaining active neurons in the network after each pruning decision used to generate Figure 4. Again, this is intended to account for the effects of cancelling interactions between neurons.

It is clear that it becomes harder to remove neurons 1-by-1 with a deeper network (which makes sense because the neurons have more interdependencies in a deeper network), but we see an overall better performance with 2nd order method vs. 1st order, except for the first 20% of the neurons (but this doesn't seem to make much difference for classification accuracy.)

Perhaps a more important observation here is that even with a more complex network, it is possible to remove up to 40% of the neurons with no major loss in performance which is clearly illustrated by the brute force curve. This shows the clear potential of an ideal pruning technique and also shows how inconsistent 1st and 2nd order Taylor Series approximations of the error can be as ranking criteria.

### 4.4.3   VISUALIZATION OF ERROR SURFACE & PRUNING DECISIONS

As seen in the case of a single layered network, these graphs are a visualization the error surface of the network output with respect to the neurons chosen for removal using each algorithm, represented in intervals of 10 neurons.

### 4.4.4   VISUALIZATION OF BRUTE FORCE PRUNING DECISIONS

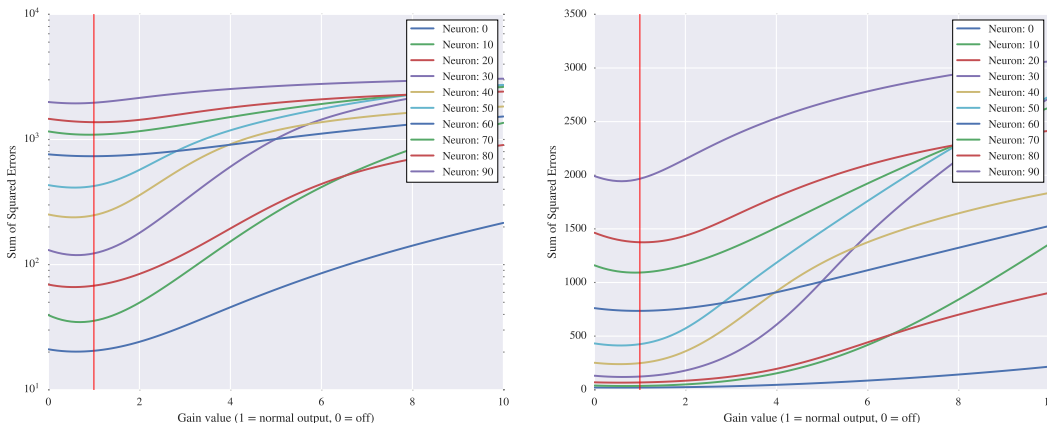

Figure 10: Error surface of the network output in log space (left) and real space (right) with respect to each candidate neuron chosen for removal using the brute force criterion; (**Network:** 2 layers, 50 neurons/layer, 10 outputs, logistic sigmoid activation, starting test accuracy: 1.000)

In Figure 10, it is clear why these neurons got chosen, their graphs clearly show little change when neuron is removed, are mostly near the floor, and show convex behaviour of error surface, which argues for the rationalization of using 2nd order methods to estimate difference in error when they are turned off.

### 4.4.5   VISUALIZATION OF 1ST ORDER APPROXIMATION PRUNING DECISIONS

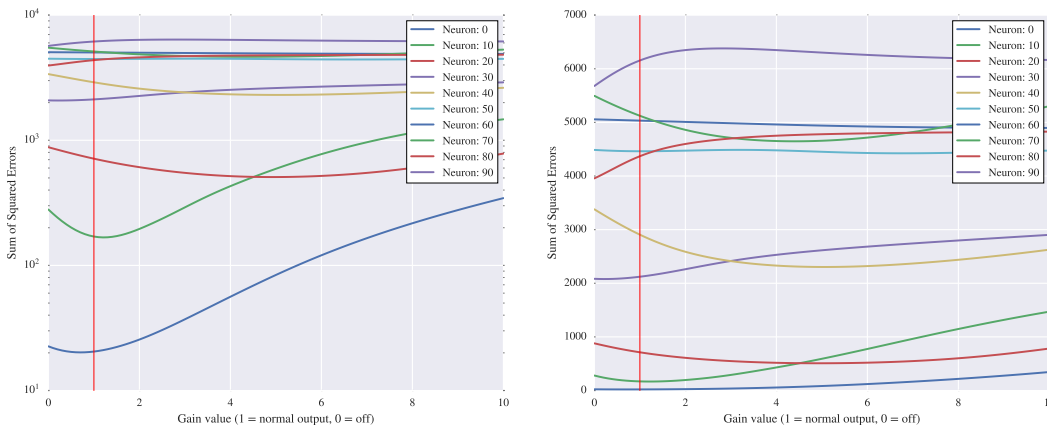

Figure 11: Error surface of the network output in log space (left) and real space (right) with respect to each candidate neuron chosen for removal using the 1st order Taylor Series error approximation criterion; (**Network:** 2 layers, 50 neurons/layer, 10 outputs, logistic sigmoid activation, starting test accuracy: 1.000)

Drawing a flat line at the point of each neurons intersection with the red vertical line (no change in gain) shows that the 1st derivative method is actually accurate for estimation of change in error in these cases, but still ultimately leads to poor decisions.

### 4.4.6 VISUALIZATION OF 2ND ORDER APPROXIMATION PRUNING DECISIONS

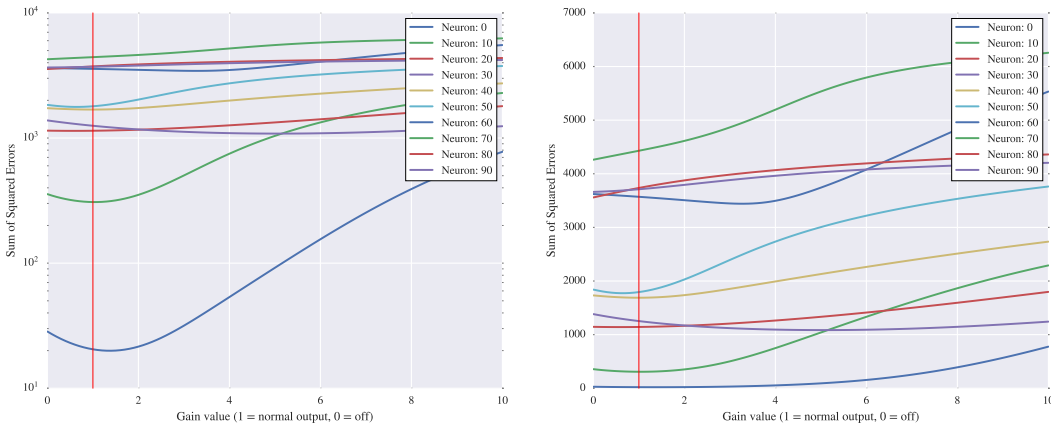

Figure 12: Error surface of the network output in log space (left) and real space (right) with respect to each candidate neuron chosen for removal using the 2nd order Taylor Series error approximation criterion; (**Network:** 2 layers, 50 neurons/layer, 10 outputs, logistic sigmoid activation, starting test accuracy: 1.000)

Clearly these neurons are not overtly poor candidates for removal (error doesn't change much between 1.0 & zero-crossing left-hand-side), but could be better (as described above in the brute force Criterion discussion).

### 4.5 INVESTIGATION OF PRUNING PERFORMANCE WITH IMPERFECT STARTING CONDITIONS

In our experiments thus far we have tacitly assumed that we start with a network which has learned an "optimal" representation of the training objective, i.e. it has been trained to the point where we accept its performance on the test set. Here we explore what happens when we prune with a sub-optimal starting network.

If the assumptions of this paper regarding the nature of neural network learning are correct, we expect that two processes are essentially at work during back-propagation training. First, we expect that the neurons which directly participate in the fundamental learning representation (even if redundantly) work together to reduce error on the training data. Second, we expect that neurons which do not directly participate in the learning representation work to cancel each other's negative influence. Furthermore, we expect that these two groups are essentially distinct, as evinced by the fact that multiple neurons can often be removed as a group with little to no effect on the network output. Some non-trivial portion of the training time, then, is spent doing work which has nothing intrinsically to do with the learning representation and essentially functions as noise cancellation.

If this is the case, when we attempt to prune a network which has not fully canceled the noisy influence of extraneous or redundant units, we might expect to see the error actually *improve* after removing a few bad apples. This is in fact what we observe, as demonstrated in the following experiments.

For each experiment in this section we trained with the full MNIST training set (LeCun & Cortes (2010)), uncompressed and without any data normalization. We trained three different networks to learn to distinguish a single handwritten digit from the rest of the data. The network architectures were each composed of 784 inputs, 1 hidden layer with 100 neurons, and 2 soft-max outputs; one to say yes, and the other to say no. These networks were trained to distinguish the digits 0, 1, and 2, and their respective starting accuracies were a sub-optimal 0.9757, 0.9881, and 0.9513. Finally, we only consider the iterative re-ranking algorithm, as the single overall ranking algorithm is clearly nonviable.

### 4.5.1   MNIST SINGLE DIGIT CLASSIFICATION: DIGIT 0

Figure 13 shows the degradation in squared error after removing neurons from a network trained to distinguish the digit 0. What we observe is that the first and second order methods both fail in different ways, though clearly the second order method makes better decisions overall. The first order method explodes spectacularly in the first few iterations. The brute force method, in stark contrast, actually *improves* in the first few iterations, and remains essentially flat until around the 60% mark, at which point it begins to gradually increase and meet the other curves.

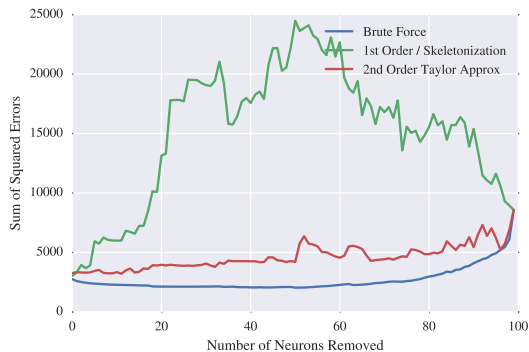

Figure 13: Degradation in squared error after pruning a single-layer network trained to do a one-versus-all classification of the digit 0 using the iterative re-ranking algorithm

The behavior of the brute force method here demonstrates that the network was essentially working to cancel the effect of a few bad neurons when the training convergence criteria were met, i.e. the network was no longer able to make progress on the training set. After removing these neurons during pruning, the output improved. We can investigate this by looking at the error surface with respect to the neurons chosen for removal by each method in turn. Below in Figure 14 is the graph of the brute force method.

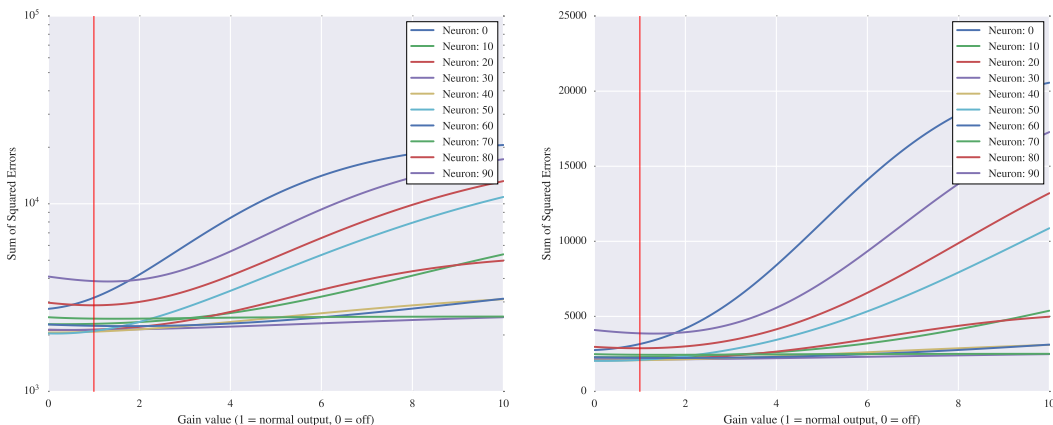

Figure 14: Error surface of the network output in log space (left) and real space (right) with respect to each candidate neuron chosen for removal using the brute force iterative re-ranking removal criterion

Figure 14 shows an interesting phenomenon, which we will see in later experiments as well. The high blue curve corresponding to neuron 0 is negatively sloped in the beginning and clearly after removing this neuron, the output will improve. The rest of the curves, in correspondence with the squared error degradation curve above, are mostly flat and tightly layered together, indicating that they are good neurons to remove.

In Figure 15 below, we observe a stark contrast to this. The curves corresponding to neurons 0 and 10 are mostly flat, and fairly lower than the rest, though clearly a mistake was made early on and the rest of the curves are clearly bad choices. In all of these cases however, we see that the curves are

easily approximated with a straight line and so the first order method may have been fairly accurate in its predictions, even though it still made poor decisions.

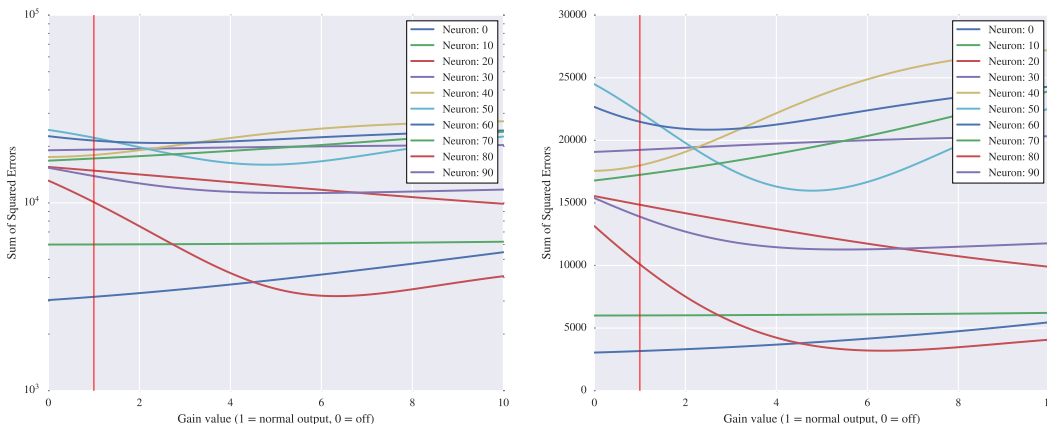

Figure 15: Error surface of the network output in log space (left) and real space (right) with respect to each candidate neuron chosen for removal using the first-order iterative re-ranking removal criterion

Figure 15 is an example of how things can go south once a few bad mistakes are made at the outset. Figure 16 shows a much better set of choices made by the second order method, though clearly not as good as the brute force method. The log-space plots make it a bit easier to see the difference between the brute force and second order methods in Figures 14 and 16, respectively.

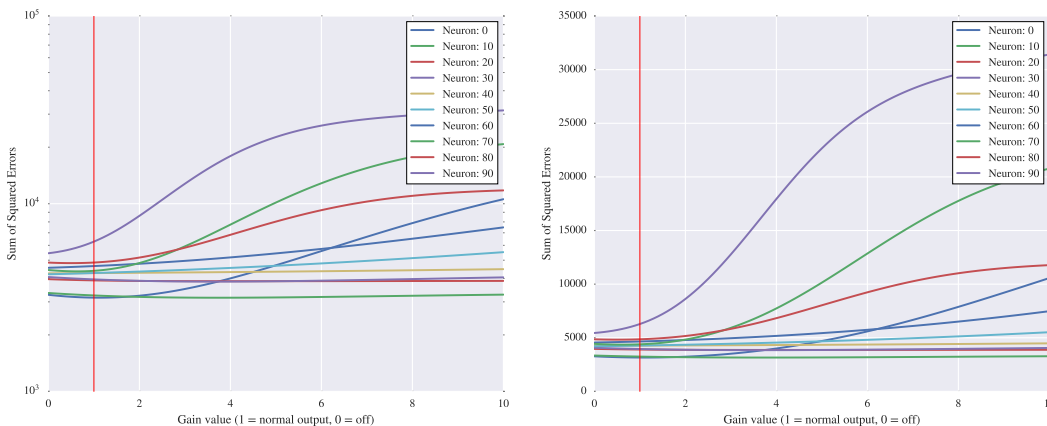

Figure 16: Error surface of the network output in log space (left) and real space (right) with respect to each candidate neuron chosen for removal using the second-order iterative re-ranking removal criterion

### 4.5.2 MNIST SINGLE DIGIT CLASSIFICATION: DIGIT 1

Examining Figure 17, we see a much starker example of the previous phenomenon, in which the brute force method continues to *improve* the performance of the network after removing 80% of the neurons in the network. The first and second order methods fail early and proceed in fits and starts (clearly demonstrating evidence of interrelated groups of noise-canceling neurons), and never fully recover. It should be noted that it would be impossible to see curves like this if neural networks evenly distributed the learning representation evenly or equitably over their hidden units.

One of the most striking things about the blue curve in Figure 17 is the fact that the network never drops below its starting error until it crosses the 80% mark, indicating that only 20% of the neurons in this network are actually essential to the learning the training objective. In this sense, we can only

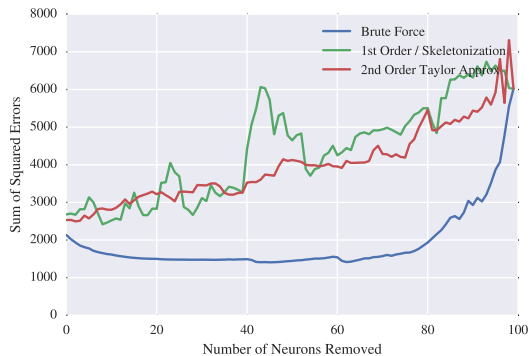

Figure 17: Degradation in squared error after pruning a single-layer network trained to do a one-versus-all classification of the digit 1 using the iterative re-ranking algorithm

wonder how much of the training time was spent winnowing the error out of the remaining 80% of the network.

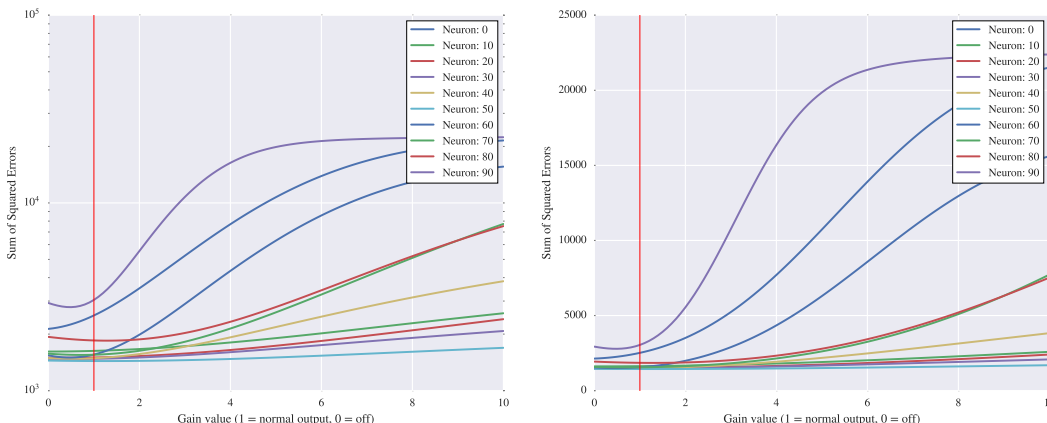

Figure 18: Error surface of the network output in log space (left) and real space (right) with respect to each candidate neuron chosen for removal using the brute force iterative re-ranking removal criterion

In Figures 18, 19 and 20 we can examine the choices made by the respective methods. The brute force method serves as our example of a near-optimal pruning regimen, and the rest are first and second order approximations of this. Small differences, clearly, can lead to large effects on the network output as shown in Figure 17.

### 4.5.3 MNIST SINGLE DIGIT CLASSIFICATION: DIGIT 2

Figure 21 is an interesting case because it shatters our confidence in the reliability of the second order method to make good pruning decisions, and further demonstrates the phenomenon of how much the error can improve if the right neurons are removed after training gets stuck. In this case, though still a poor performance overall, the first order method vastly outperforms the second order method.

Figure 22 shows us a clear example of the first element to remove having a negative error slope, and improving the output as a result. The rest of the pruning decisions are reasonable. Comparing with the blue curve in Figure 21, we see the correspondence between the first pruning decision improving the output, and the remaining pruning decisions keeping the output fairly flat. Clearly, however, there isn't much room to get worse given our starting point with a sub-optimal network, and we see that the ending sum of squared errors is not much higher than the starting point. At the same time, we can still see the contrast in performance if we make optimal pruning decisions, and most of the neurons in this network were clearly doing nothing.

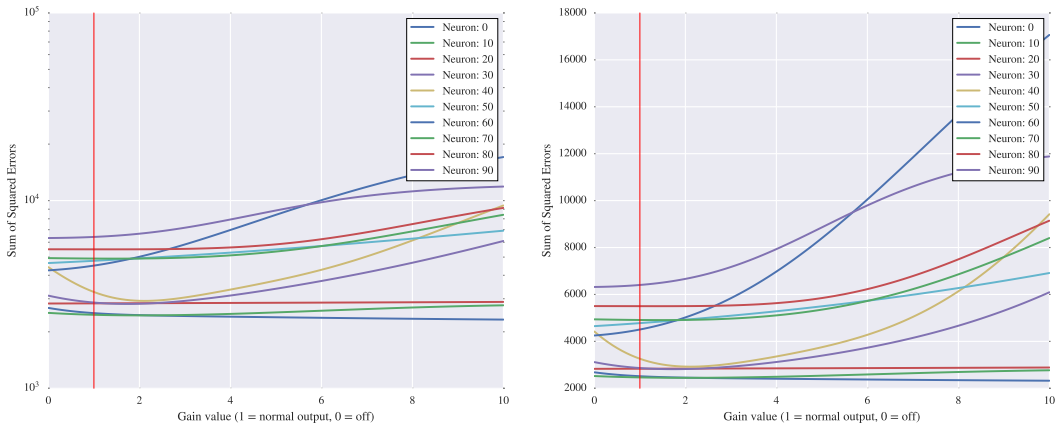

Figure 19: Error surface of the network output in log space (left) and real space (right) with respect to each candidate neuron chosen for removal using the first-order iterative re-ranking removal criterion

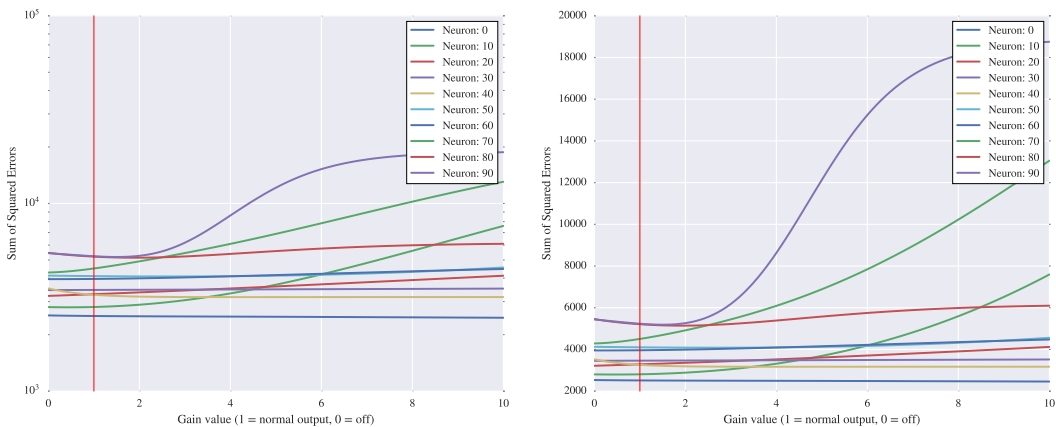

Figure 20: Error surface of the network output in log space (left) and real space (right) with respect to each candidate neuron chosen for removal using the second-order iterative re-ranking removal criterion

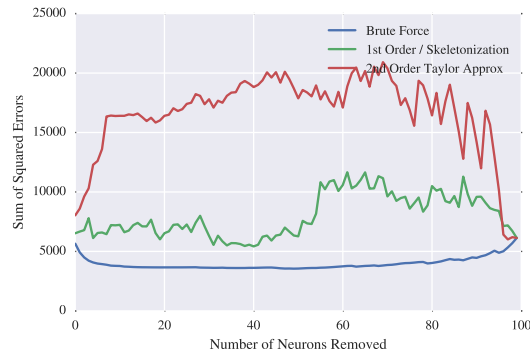

Figure 21: Degradation in squared error after pruning a single-layer network trained to do a one-versus-all classification of the digit 2 using the iterative re-ranking algorithm

In Figure 23, we see a mixed bag in which the decisions are clearly sub-optimal, though much better than Figure 24, in which we can observe how a bad first decision essentially ruined the network for good. The jagged edges of the red curve in Figure 21 correspond with the positive and negative

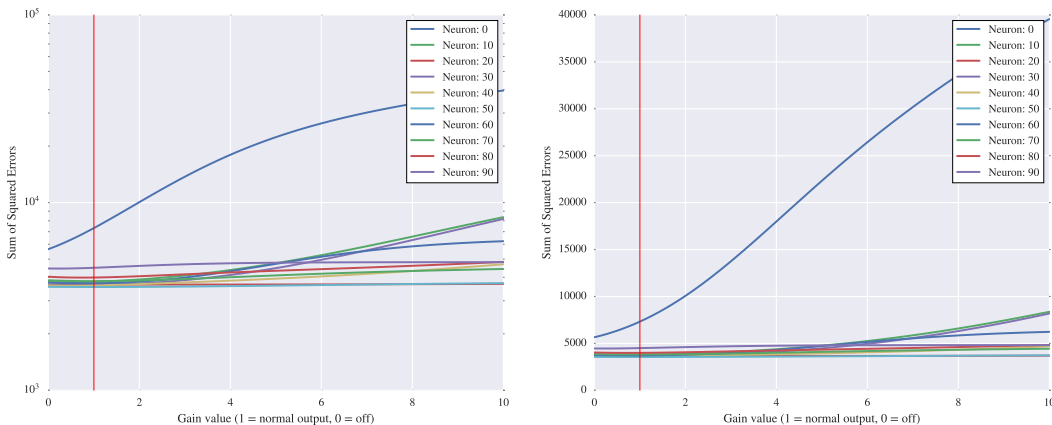

Figure 22: Error surface of the network output in log space (left) and real space (right) with respect to each candidate neuron chosen for removal using the brute force iterative re-ranking removal criterion

slopes of the cluster of bad pruning decisions in 24. Once again, these are not necessarily bad decisions, but the starting point is already bad and this cannot be recovered without re-training the network.

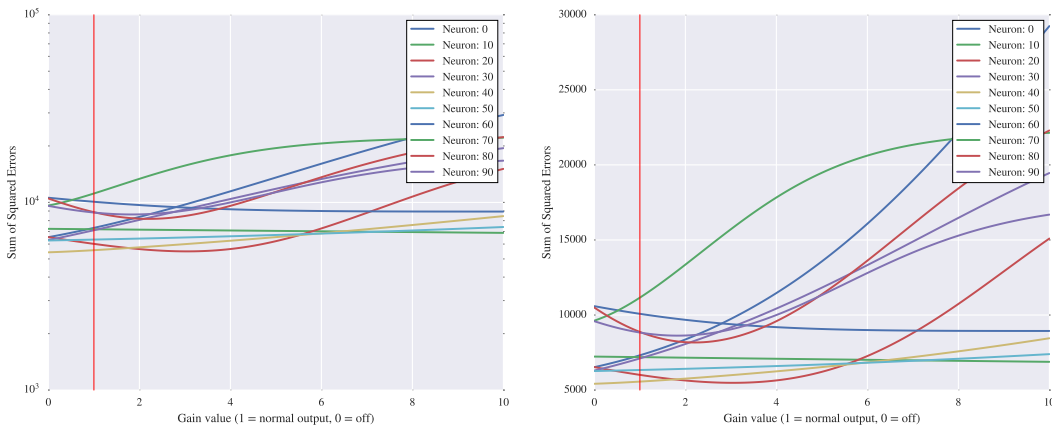

Figure 23: Error surface of the network output in log space (left) and real space (right) with respect to each candidate neuron chosen for removal using the first-order iterative re-ranking removal criterion

### 4.5.4 ASIDE: IMPLICATIONS OF THIS EXPERIMENT

From the three examples above, we see that in each case, starting from a sub-optimal network, a brute force removal technique consistently improves performance for the first few pruning iterations, and the sum of squared errors does not degrade beyond the starting point until around 60-80% of the neurons have been removed. This is only possible if we have an essentially strict dichotomy between the roles of different neurons during training. If the network needs only 20-40% of the neurons it began with, the training process is essentially dominated by the task of canceling the residual noise of redundant neurons. Furthermore, the network can get stuck in training with redundant units and distort the final output. This is strong evidence of our thesis that the learning representation is neither equitably nor evenly distributed and that most of the neurons which do not directly participate in the learning representation can be removed without any retraining.

### 4.6 EXPERIMENTS ON TOY DATASETS

As can be seen from the experiments on MNIST, even though the 2nd-order approximation criterion is consistently better than 1st-order, its performance is not nearly as good as brute force based rank-

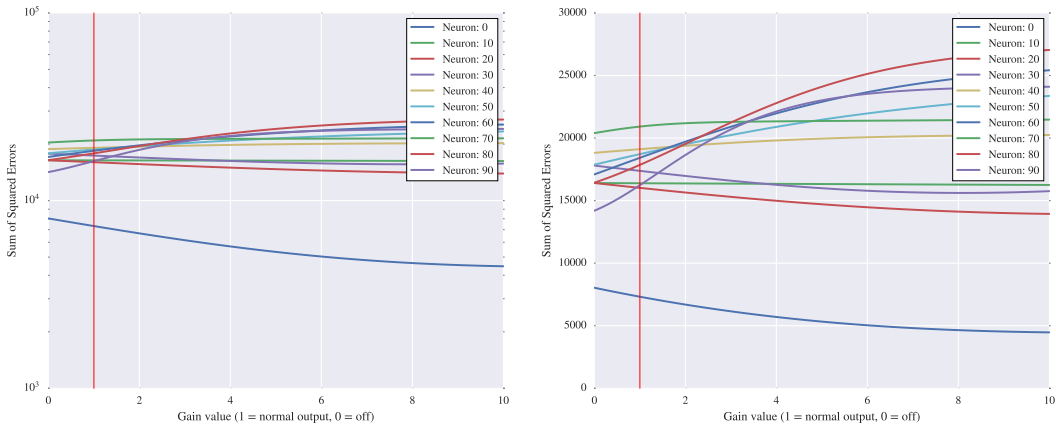

Figure 24: Error surface of the network output in log space (left) and real space (right) with respect to each candidate neuron chosen for removal using the second-order iterative re-ranking removal criterion

ing, especially beyond the first layer. What is interesting to note is that from some other experiments conducted on toy datasets (predicting whether a given point would lie inside a given shape on the Cartesian plane), the performance of the 2nd-order method was found to be exceptionally good and produced results very close to the brute force method. The 1st-order method, as expected, performed poorly here as well. Some of these results are illustrated in Figure 25.

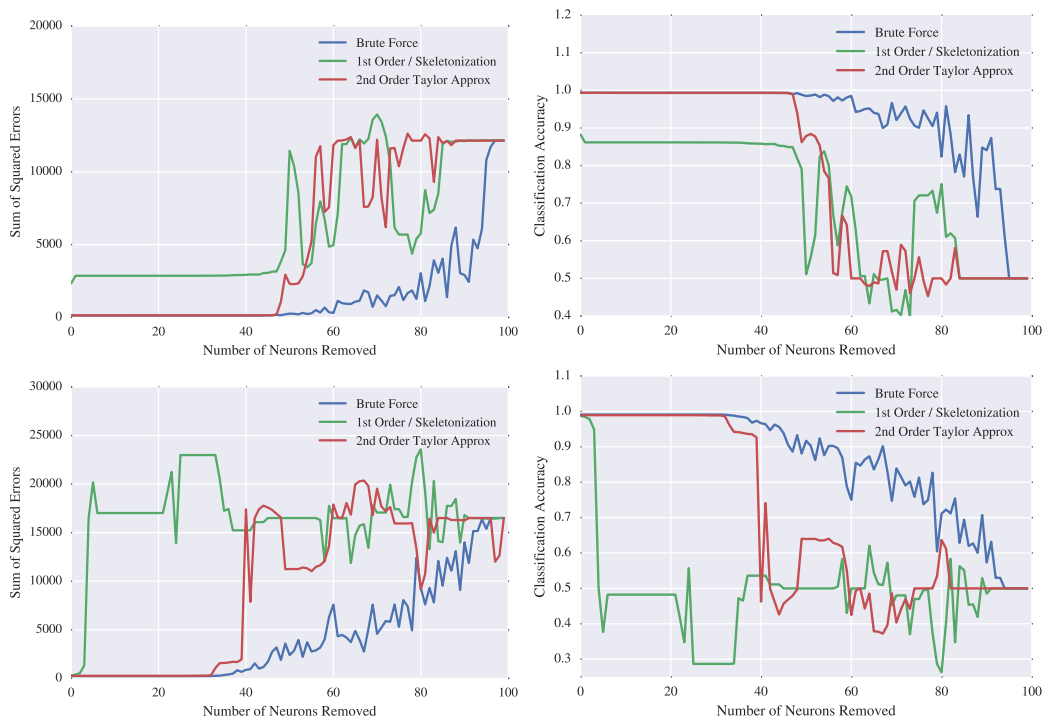

Figure 25: Degradation in squared error (left) and classification accuracy (right) after pruning a 2-layer network using the iterative re-ranking algorithm on a toy "diamond" shape dataset (top) and a toy "random shape" dataset (below); (**Network:** 2 layers, 50 neurons/layer, 10 outputs, logistic sigmoid activation, starting test accuracy: 0.992(diamond); 0.986(random shape)

## 5 Conclusions & Future Work

In conclusion, we must first re-assert that we do not present this work as a bench-marking study of the algorithm we derived and tested. We have merely used this algorithm as a jumping off point to investigate the nature of learning representations in neural networks. What we discovered is that first and second order methods do not make particularly good pruning decisions, and can get hopelessly lost after making a bad pruning decision resulting in a network fault. Furthermore, the brute-force algorithm does surprisingly well, despite being computationally expensive. This method does *so* well in fact, we argue that further investigation is warranted to make this algorithm computationally tractable, though we do not speculate on how that should be done here.

We also observed strong evidence for the hypotheses of Mozer & Smolensky (1989a) regarding the "dualist" nature of hidden units, i.e. that learning representations are divided between units which either participate in the output approximation or learn to cancel each others influence. This suggests that neural networks may in fact learn a minimal network implicitly, though we cannot say for sure that this is the case without further investigation. A necessary experiment to this end would be to compare the size of network constructed using cascade correlation (Fahlman & Lebiere (1989)) and compare it to the results described herein.

We have presented a novel algorithm for pruning whole neurons from a trained neural network using a second-order Taylor series approximation of the change in error resulting from the removal a given neuron as a pruning criteria. We compared this method to a first order method and a brute-force serial removal method which exhaustively found the next best single neuron to remove at each stage. Our algorithm relies on a combination of assumptions similar to the ones made by Mozer & Smolensky (1989a) and LeCun et al. (1989) in the formulation of the Skeletonization and Optimal Brain Damage algorithms.

First, we assumed that the error function with respect to each individual neuron can be approximated with a straight line or more precisely with a parabola. Second, for second derivative terms we consider only the diagonal elements of the Hessian matrix, i.e. we assume that each neuron-weight connection can be treated independently of the other elements in the network. Third, we assumed that pruning could be done in a serial fashion in which we find the single least productive element in the network, remove it, and move on. We found that all of these assumptions are deeply flawed in the sense that the true relevance of a neuron can only be partially approximated by a first or second order method, and only at certain stages of the pruning process.

For most problems, these methods can usually remove between 10-30% of the neurons in a trained network, but beyond this point their reliability breaks down. For certain problems, none of the described methods seem to perform very well, though for obvious reasons the brute-force method always exhibits the best results. The reason for this is that the error function with respect to each hidden unit is more complex than a simple second-order Taylor series can approximate. Furthermore, we have not directly taken into account the interdependence of elements within a network, though the work of Hassibi & Stork (1993) could provide some guidance in this regard. This is another critical issue to investigate in the future.

Re-training may help in this regard. We freely admit that our algorithm does not use re-training to recover from errors made in pruning decisions. We argue that evaluating a network pruning algorithm using re-training does not allow us to make fair comparisons between the kinds of decisions made by these algorithms. Neural networks are very good at recovering from the removal of individual elements with re-training and so this compensates for sub-optimal pruning criteria.

We have observed that pruning whole neurons from an optimally trained network without major loss in performance is not only possible but also enables compressing networks to 40-70% of their original size, which is of great importance in constrained memory environments like embedded devices. We cite the results of our experiments using the brute force criterion as evidence of this conclusion. However expensive, it would be extremely easy to parallelize this method, or potentially approximate it using a subset of the training data to decide which neurons to prune. This avoids the problem of trying to approximate the importance of a unit and potentially making a mistake.

It would also be interesting to see how these methods perform on deeper networks and on some other popular and real world datasets. In our case, on the MNIST dataset, we observed that it was more difficult to prune neurons from a deeper network than from one with a single layer. We should expect

this trend to continue as networks get deeper and deeper, which also calls into further question the reliability of the described first and second order methods. We did investigate the order in which neurons were plucked from each layer of the networks and we found that the brute force method primarily removes neurons from the deepest layer of the network first, but there was no obvious pattern in layer preference for the other two methods.

Our experiments using the visualization of error surfaces and pruning decisions concretely establish the fact that not all neurons in a network contribute to its performance in the same way, and the observed complexity of these functions demonstrates limitations of the approximations we used.

Finally, we encourage the readers of this work to take these results into consideration when making decisions as to which methods to use to improve network generalization or compress their models. It should be remembered that various heuristics may perform well in practice for reasons which are in fact orthogonal to the accepted justifications given by their proponents.

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

APPENDIX

## A  SECOND DERIVATIVE BACK-PROPAGATION

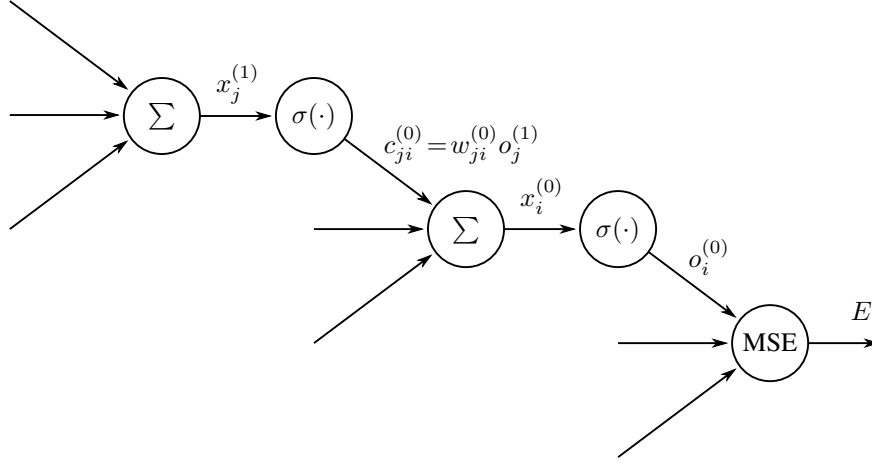

Figure 26: A computational graph of a simple feed-forward network illustrating the naming of different variables, where $\sigma(\cdot)$ is the nonlinearity, MSE is the mean-squared error cost function and $E$ is the overall loss.

Name and network definitions:

$$E = \frac{1}{2}\sum_i (o_i^{(0)} - t_i)^2 \quad o_i^{(m)} = \sigma(x_i^{(m)}) \quad x_i^{(m)} = \sum_j w_{ji}^{(m)} o_j^{(m+1)} \quad c_{ji}^{(m)} = w_{ji}^{(m)} o_j^{(m+1)} \quad (5)$$

Superscripts represent the index of the layer of the network in question, with 0 representing the output layer. $E$ is the squared-error network cost function. $o_i^{(m)}$ is the $i$th output in layer $m$ generated by the activation function $\sigma$, which in this paper is is the standard logistic sigmoid. $x_i^{(m)}$ is the weighted sum of inputs to the $i$th neuron in the $m$th layer, and $c_{ji}^{(m)}$ is the contribution of the $j$th neuron in the $m+1$ layer to the input of the $i$th neuron in the $m$th layer.

### A.1  FIRST AND SECOND DERIVATIVES

The first and second derivatives of the cost function with respect to the outputs:

$$\frac{\partial E}{\partial o_i^{(0)}} = o_i^{(0)} - t_i \qquad (6)$$

$$\frac{\partial^2 E}{\partial o_i^{(0)^2}} = 1 \qquad (7)$$

The first and second derivatives of the sigmoid function in forms depending only on the output:

$$\sigma'(x) = \sigma(x)\,(1 - \sigma(x)) \qquad (8)$$
$$\sigma''(x) = \sigma'(x)\,(1 - 2\sigma(x)) \qquad (9)$$

The second derivative of the sigmoid is easily derived from the first derivative:

$$\sigma'(x) = \sigma(x)\left(1 - \sigma(x)\right) \tag{10}$$

$$\sigma''(x) = \frac{\mathrm{d}}{\mathrm{d}x} \underbrace{\sigma(x)}_{f(x)} \underbrace{\left(1 - \sigma(x)\right)}_{g(x)} \tag{11}$$

$$\sigma''(x) = f'(x)g(x) + f(x)g'(x) \tag{12}$$

$$\sigma''(x) = \sigma'(x)(1 - \sigma(x)) - \sigma(x)\sigma'(x) \tag{13}$$

$$\sigma''(x) = \sigma'(x) - 2\sigma(x)\sigma'(x) \tag{14}$$

$$\sigma''(x) = \sigma'(x)(1 - 2\sigma(x)) \tag{15}$$

And for future convenience:

$$\frac{\mathrm{d}o_i^{(m)}}{\mathrm{d}x_i^{(m)}} = \frac{\mathrm{d}}{\mathrm{d}x_i^{(m)}}\left(o_i^{(m)} = \sigma(x_i^{(m)})\right) \tag{16}$$

$$= \left(o_i^{(m)}\right)\left(1 - o_i^{(m)}\right) \tag{17}$$

$$= \sigma'\left(x_i^{(m)}\right) \tag{18}$$

$$\frac{\mathrm{d}^2 o_i^{(m)}}{\mathrm{d}x_i^{(m)2}} = \frac{\mathrm{d}}{\mathrm{d}x_i^{(m)}}\left(\frac{\mathrm{d}o_i^{(m)}}{\mathrm{d}x_i^{(m)}} = \left(o_i^{(m)}\right)\left(1 - o_i^{(m)}\right)\right) \tag{19}$$

$$= \left(o_i^{(m)}\left(1 - o_i^{(m)}\right)\right)\left(1 - 2o_i^{(m)}\right) \tag{20}$$

$$= \sigma''\left(x_i^{(m)}\right) \tag{21}$$

Derivative of the error with respect to the $i$th neuron's input $x_i^{(0)}$ in the output layer:

$$\frac{\partial E}{\partial x_i^{(0)}} = \frac{\partial E}{\partial o_i^{(0)}}\frac{\partial o_i^{(0)}}{\partial x_i^{(0)}} \tag{22}$$

$$= \underbrace{\left(o_i^{(0)} - t_i\right)}_{\text{from (6)}} \underbrace{\sigma\left(x_i^{(0)}\right)\left(1 - \sigma\left(x_i^{(0)}\right)\right)}_{\text{from (8)}} \tag{23}$$

$$= \left(o_i^{(0)} - t_i\right)\left(o_i^{(0)}\left(1 - o_i^{(0)}\right)\right) \tag{24}$$

$$= \left(o_i^{(0)} - t_i\right)\sigma'\left(x_i^{(0)}\right) \tag{25}$$

Second derivative of the error with respect to the $i$th neuron's input $x_i^{(0)}$ in the output layer:

$$\frac{\partial^2 E}{\partial x_i^{(0)2}} = \frac{\partial}{\partial x_i^{(0)}} \left( \frac{\partial E}{\partial o_i^{(0)}} \frac{\partial o_i^{(0)}}{\partial x_i^{(0)}} \right) \tag{26}$$

$$= \frac{\partial^2 E}{\partial x_i^{(0)} \partial o_i^{(0)}} \frac{\partial o_i^{(0)}}{\partial x_i^{(0)}} + \frac{\partial E}{\partial o_i^{(0)}} \frac{\partial^2 o_i^{(0)}}{\partial x_i^{(0)2}} \tag{27}$$

$$= \frac{\partial^2 E}{\partial x_i^{(0)} \partial o_i^{(0)}} \underbrace{\left( o_i^{(0)} \left( 1 - o_i^{(0)} \right) \right)}_{\text{from (8)}} + \underbrace{\left( o_i^{(0)} - t_i \right)}_{\text{from (6)}} \underbrace{\left( o_i^{(0)} \left( 1 - o_i^{(0)} \right) \right) \left( 1 - 2o_i^{(0)} \right)}_{\text{from (9)}} \tag{28}$$

$$\left( \frac{\partial^2 E}{\partial x_i^{(0)} \partial o_i^{(0)}} \right) = \frac{\partial}{\partial x_i^{(0)}} \frac{\partial E}{\partial o_i^{(0)}} = \frac{\partial}{\partial x_i^{(0)}} \underbrace{\left( o_i^{(0)} - t_i \right)}_{\text{from (6)}} = \frac{\partial o_i^{(0)}}{\partial x_i^{(0)}} = \underbrace{\left( o_i^{(0)} \left( 1 - o_i^{(0)} \right) \right)}_{\text{from (8)}} \tag{29}$$

$$\frac{\partial^2 E}{\partial x_i^{(0)2}} = \left( o_i^{(0)} \left( 1 - o_i^{(0)} \right) \right)^2 + \left( o_i^{(0)} - t_i \right) \left( o_i^{(0)} \left( 1 - o_i^{(0)} \right) \right) \left( 1 - 2o_i^{(0)} \right) \tag{30}$$

$$= \left( \sigma' \left( x_i^{(0)} \right) \right)^2 + \left( o_i^{(0)} - t_i \right) \sigma'' \left( x_i^{(0)} \right) \tag{31}$$

First derivative of the error with respect to a single input contribution $c_{ji}^{(0)}$ from neuron $j$ to neuron $i$ with weight $w_{ji}^{(0)}$ in the output layer:

$$\frac{\partial E}{\partial c_{ji}^{(0)}} = \frac{\partial E}{\partial o_i^{(0)}} \frac{\partial o_i^{(0)}}{\partial x_i^{(0)}} \frac{\partial x_i^{(0)}}{\partial c_{ji}^{(0)}} \tag{32}$$

$$= \underbrace{\left( o_i^{(0)} - t_i \right)}_{\text{from (6)}} \underbrace{\left( o_i^{(0)} \left( 1 - o_i^{(0)} \right) \right)}_{\text{from (8)}} \frac{\partial x_i^{(0)}}{\partial c_{ji}^{(0)}} \tag{33}$$

$$\left( \frac{\partial x_i^{(m)}}{\partial c_{ji}^{(m)}} \right) = \frac{\partial}{\partial c_{ji}^{(m)}} \left( x_i^{(m)} = \sum_j w_{ji}^{(m)} o_j^{(m+1)} \right) = \frac{\partial}{\partial c_{ji}^{(m)}} \left( c_{ji}^{(m)} + k \right) = 1 \tag{34}$$

$$\frac{\partial E}{\partial c_{ji}^{(0)}} = \left( o_i^{(0)} - t_i \right) \left( o_i^{(0)} \left( 1 - o_i^{(0)} \right) \right) \tag{35}$$

$$= \underbrace{\left( o_i^{(0)} - t_i \right) \sigma' \left( x_i^{(0)} \right)}_{\text{from (25)}} \tag{36}$$

$$\frac{\partial E}{\partial c_{ji}^{(0)}} = \frac{\partial E}{\partial x_i^{(0)}} \tag{37}$$

Second derivative of the error with respect to a single input contribution $c_{ji}^{(0)}$:

$$\frac{\partial^2 E}{\partial c_{ji}^{(0)2}} = \frac{\partial}{\partial c_{ji}^{(0)}} \left( \frac{\partial E}{\partial c_{ji}^{(0)}} = \underbrace{\left(o_i^{(0)} - t_i\right) \sigma'\left(x_i^{(0)}\right)}_{\text{from (36)}} \right) \tag{38}$$

$$= \frac{\partial}{\partial c_{ji}^{(0)}} \left( \sigma\left(x_i^{(0)}\right) - t_i \right) \sigma'\left(x_i^{(0)}\right) \tag{39}$$

$$= \frac{\partial}{\partial c_{ji}^{(0)}} \left( \sigma\left(\sum_j w_{ji}^{(m)} o_j^{(m+1)}\right) - t_i \right) \sigma'\left(\sum_j w_{ji}^{(m)} o_j^{(m+1)}\right) \tag{40}$$

$$= \frac{\partial}{\partial c_{ji}^{(0)}} \left( \sigma\left(\sum_j c_{ji}^{(0)}\right) - t_i \right) \sigma'\left(\sum_j c_{ji}^{(0)}\right) \tag{41}$$

$$= \frac{\partial}{\partial c_{ji}^{(0)}} \underbrace{\left( \sigma\left(c_{ji}^{(0)} + k\right) - t_i \right)}_{f\left(c_{ji}^{(0)}\right)} \underbrace{\sigma'\left(c_{ji}^{(0)} + k\right)}_{g\left(c_{ji}^{(0)}\right)} \tag{42}$$

We now make use of the abbreviations $f$ and $g$:

$$= f'\left(c_{ji}^{(0)}\right) g\left(c_{ji}^{(0)}\right) + f\left(c_{ji}^{(0)}\right) g'\left(c_{ji}^{(0)}\right) \tag{43}$$

$$= \sigma'\left(c_{ji}^{(0)} + k\right) \sigma'\left(c_{ji}^{(0)} + k\right) + \left( \sigma\left(c_{ji}^{(0)} + k\right) - t_i \right) \sigma''\left(c_{ji}^{(0)} + k\right) \tag{44}$$

$$= \sigma'\left(c_{ji}^{(0)} + k\right)^2 + \left(o_i^{(0)} - t_i\right) \sigma''\left(c_{ji}^{(0)} + k\right) \tag{45}$$

$$\left( c_{ji}^{(0)} + k = \sum_j c_{ji}^{(0)} = \sum_j w_{ji}^{(m)} o_j^{(m+1)} = x_i^{(0)} \right) \tag{46}$$

$$\frac{\partial^2 E}{\partial c_{ji}^{(0)2}} = \underbrace{\left( \sigma'\left(x_i^{(0)}\right) \right)^2 + \left(o_i^{(0)} - t_i\right) \sigma''\left(x_i^{(0)}\right)}_{\text{from (31)}} \tag{47}$$

$$\frac{\partial^2 E}{\partial c_{ji}^{(0)2}} = \frac{\partial^2 E}{\partial x_i^{(0)2}} \tag{48}$$

### A.1.1 SUMMARY OF OUTPUT LAYER DERIVATIVES

$$\frac{\partial E}{\partial o_i^{(0)}} = o_i^{(0)} - t_i \qquad\qquad \frac{\partial^2 E}{\partial o_i^{(0)2}} = 1 \tag{49}$$

$$\frac{\partial E}{\partial x_i^{(0)}} = \left(o_i^{(0)} - t_i\right) \sigma'\left(x_i^{(0)}\right) \qquad \frac{\partial^2 E}{\partial x_i^{(0)2}} = \left( \sigma'\left(x_i^{(0)}\right) \right)^2 + \left(o_i^{(0)} - t_i\right) \sigma''\left(x_i^{(0)}\right) \tag{50}$$

$$\frac{\partial E}{\partial c_{ji}^{(0)}} = \frac{\partial E}{\partial x_i^{(0)}} \qquad\qquad \frac{\partial^2 E}{\partial c_{ji}^{(0)2}} = \frac{\partial^2 E}{\partial x_i^{(0)2}} \tag{51}$$

### A.1.2 Hidden Layer Derivatives

The first derivative of the error with respect to a neuron with output $o_j^{(1)}$ in the first hidden layer, summing over all partial derivative contributions from the output layer:

$$\frac{\partial E}{\partial o_j^{(1)}} = \sum_i \frac{\partial E}{\partial o_i^{(0)}} \frac{\partial o_i^{(0)}}{\partial x_i^{(0)}} \frac{\partial x_i^{(0)}}{\partial c_{ji}^{(0)}} \frac{\partial c_{ji}^{(0)}}{\partial o_j^{(1)}} = \sum_i \underbrace{\left(o_i^{(0)} - t_i\right) \sigma'\left(x_i^{(0)}\right)}_{\text{from (25)}} w_{ji}^{(0)} \tag{52}$$

$$\frac{\partial c_{ji}^{(m)}}{\partial o_j^{(m+1)}} = \frac{\partial}{\partial o_j^{(m+1)}} \left(c_{ji}^{(m)} = w_{ji}^{(m)} o_j^{(m+1)}\right) = w_{ji}^{(m)} \tag{53}$$

$$\frac{\partial E}{\partial o_j^{(1)}} = \sum_i \frac{\partial E}{\partial x_i^{(0)}} w_{ji}^{(0)} \tag{54}$$

Note that this equation does not depend on the specific form of $\frac{\partial E}{\partial x_i^{(0)}}$, whether it involves a sigmoid or any other activation function. We can therefore replace the specific indexes with general ones, and use this equation in the future.

$$\frac{\partial E}{\partial o_j^{(m+1)}} = \sum_i \frac{\partial E}{\partial x_i^{(m)}} w_{ji}^{(m)} \tag{55}$$

The second derivative of the error with respect to a neuron with output $o_j^{(1)}$ in the first hidden layer:

$$\frac{\partial^2 E}{\partial o_j^{(1)^2}} = \frac{\partial}{\partial o_j^{(1)}} \frac{\partial E}{\partial o_j^{(1)}} \tag{56}$$

$$= \frac{\partial}{\partial o_j^{(1)}} \sum_i \frac{\partial E}{\partial x_i^{(0)}} w_{ji}^{(0)} \tag{57}$$

$$= \frac{\partial}{\partial o_j^{(1)}} \sum_i \left(o_i^{(0)} - t_i\right) \sigma'\left(x_i^{(0)}\right) w_{ji}^{(0)} \tag{58}$$

If we now make use of the fact, that $o_i^{(0)} = \sigma\left(x_i^{(0)}\right) = \sigma\left(\sum_j \left(w_{ji}^{(0)} o_j^{(1)}\right)\right)$, we can evaluate the expression further.

$$\frac{\partial^2 E}{\partial o_j^{(1)^2}} = \frac{\partial}{\partial o_j^{(1)}} \sum_i \underbrace{\left(\sigma\left(\sum_j w_{ji}^{(0)} o_j^{(1)}\right) - t_i\right)}_{f\left(o_j^{(1)}\right)} \underbrace{\sigma'\left(\sum_j w_{ji}^{(0)} o_j^{(1)}\right) w_{ji}^{(0)}}_{g\left(o_j^{(1)}\right)} \tag{59}$$

$$= \sum_i \left(f'\left(o_j^{(1)}\right) g\left(o_j^{(1)}\right) + f\left(o_j^{(1)}\right) g'\left(o_j^{(1)}\right)\right) \tag{60}$$

$$= \sum_i \sigma'\left(\sum_j w_{ji}^{(0)} o_j^{(1)}\right) w_{ji}^{(0)} \sigma'\left(\sum_j w_{ji}^{(0)} o_j^{(1)}\right) w_{ji}^{(0)} + \ldots \tag{61}$$

$$\sum_i \left(\sigma\left(\sum_j w_{ji}^{(0)} o_j^{(1)}\right) - t_i\right) \sigma''\left(\sum_j w_{ji}^{(0)} o_j^{(1)}\right) \left(w_{ji}^{(0)}\right)^2 \tag{62}$$

$$= \sum_i \left(\left(\sigma'\left(x_i^{(0)}\right)\right)^2 \left(w_{ji}^{(0)}\right)^2 + \left(o_i^{(0)} - t_i\right) \sigma''\left(x_i^{(0)}\right) \left(w_{ji}^{(0)}\right)^2\right) \tag{63}$$

$$= \sum_i \underbrace{\left(\left(\sigma'\left(x_i^{(0)}\right)\right)^2 + \left(o_i^{(0)} - t_i\right) \sigma''\left(x_i^{(0)}\right)\right)}_{\text{from (31)}} \left(w_{ji}^{(0)}\right)^2 \tag{64}$$

Summing up, we obtain the more general expression:

$$\frac{\partial^2 E}{\partial o_j^{(1)^2}} = \sum_i \frac{\partial^2 E}{\partial x_i^{(0)^2}} \left( w_{ji}^{(0)} \right)^2 \tag{65}$$

Note that the equation in (65) does not depend on the form of $\frac{\partial^2 E}{\partial x_x^{(0)^2}}$, which means we can replace the specific indexes with general ones:

$$\frac{\partial^2 E}{\partial o_j^{(m+1)^2}} = \sum_i \frac{\partial^2 E}{\partial x_i^{(m)^2}} \left( w_{ji}^{(m)} \right)^2 \tag{66}$$

At this point we are beginning to see the recursion in the form of the 2nd derivative terms which can be thought of analogously to the first derivative recursion which is central to the back-propagation algorithm. The formulation above which makes specific reference to layer indexes also works in the general case.

Consider the $i$th neuron in any layer $m$ with output $o_i^{(m)}$ and input $x_i^{(m)}$. The first and second derivatives of the error $E$ with respect to this neuron's *input* are:

$$\frac{\partial E}{\partial x_i^{(m)}} = \frac{\partial E}{\partial o_i^{(m)}} \frac{\partial o_i^{(m)}}{\partial x_i^{(m)}} \tag{67}$$

$$\frac{\partial^2 E}{\partial x_i^{(m)^2}} = \frac{\partial}{\partial x_i^{(m)}} \frac{\partial E}{\partial x_i^{(m)}} \tag{68}$$

$$= \frac{\partial}{\partial x_i^{(m)}} \left( \frac{\partial E}{\partial o_i^{(m)}} \frac{\partial o_i^{(m)}}{\partial x_i^{(m)}} \right) \tag{69}$$

$$= \frac{\partial^2 E}{\partial x_i^{(m)} \partial o_i^{(m)}} \frac{\partial o_i^{(m)}}{\partial x_i^{(m)}} + \frac{\partial E}{\partial o_i^{(m)}} \frac{\partial^2 o_i^{(m)}}{\partial x_i^{(m)^2}} \tag{70}$$

$$= \frac{\partial}{\partial o_i^{(m)}} \left( \frac{\partial E}{\partial x_i^{(m)}} = \frac{\partial E}{\partial o_i^{(m)}} \frac{\partial o_i^{(m)}}{\partial x_i^{(m)}} \right) \frac{\partial o_i^{(m)}}{\partial x_i^{(m)}} + \frac{\partial E}{\partial o_i^{(m)}} \sigma'' \left( x_i^{(m)} \right) \tag{71}$$

$$= \frac{\partial^2 E}{\partial o_i^{(m)^2}} \left( \frac{\partial o_i^{(m)}}{\partial x_i^{(m)}} \frac{\partial o_i^{(m)}}{\partial x_i^{(m)}} \right) + \frac{\partial E}{\partial o_i^{(m)}} \sigma'' \left( x_i^{(m)} \right) \tag{72}$$

$$\frac{\partial^2 E}{\partial x_i^{(m)^2}} = \frac{\partial^2 E}{\partial o_i^{(m)^2}} \left( \sigma' \left( x_i^{(m)} \right) \right)^2 + \frac{\partial E}{\partial o_i^{(m)}} \sigma'' \left( x_i^{(m)} \right) \tag{73}$$

Note the form of this equation is the general form of what was derived for the output layer in (31). Both of the above first and second terms are easily computable and can be stored as we propagate back from the output of the network to the input. With respect to the output layer, the first and second derivative terms have already been derived above. In the case of the $m + 1$ hidden layer during back propagation, there is a summation of terms calculated in the $m$th layer. For the first derivative, we have this from (55).

$$\frac{\partial E}{\partial o_j^{(m+1)}} = \sum_i \frac{\partial E}{\partial x_i^{(m)}} w_{ji}^{(m)} \tag{74}$$

And the second derivative for the $j$th neuron in the $m + 1$ layer:

$$\frac{\partial^2 E}{\partial x_j^{(m+1)^2}} = \frac{\partial^2 E}{\partial o_j^{(m+1)^2}} \left( \sigma' \left( x_j^{(m+1)} \right) \right)^2 + \frac{\partial E}{\partial o_j^{(m+1)}} \sigma'' \left( x_j^{(m+1)} \right) \tag{75}$$

We can replace both derivative terms with the forms which depend on the previous layer:

$$\frac{\partial^2 E}{\partial x_j^{(m+1)^2}} = \underbrace{\sum_i \frac{\partial^2 E}{\partial x_i^{(0)^2}} \left( w_{ji}^{(0)} \right)^2}_{\text{from } (66)} \left( \sigma' \left( x_j^{(m+1)} \right) \right)^2 + \underbrace{\sum_i \frac{\partial E}{\partial x_i^{(m)}} w_{ji}^{(m)}}_{\text{from } (55)} \sigma'' \left( x_j^{(m+1)} \right) \tag{76}$$

And this horrible mouthful of an equation gives you a general form for any neuron in the $j$th position of the $m + 1$ layer. Taking very careful note of the indexes, this can be more or less translated painlessly to code. You are welcome, world.

### A.1.3 SUMMARY OF HIDDEN LAYER DERIVATIVES

$$\frac{\partial E}{\partial o_j^{(m+1)}} = \sum_i \frac{\partial E}{\partial x_i^{(m)}} w_{ji}^{(m)} \qquad \frac{\partial^2 E}{\partial o_j^{(m+1)2}} = \sum_i \frac{\partial^2 E}{\partial x_i^{(m)2}} \left(w_{ji}^{(m)}\right)^2 \tag{77}$$

$$\frac{\partial E}{\partial x_i^{(m)}} = \frac{\partial E}{\partial o_i^{(m)}} \frac{\partial o_i^{(m)}}{\partial x_i^{(m)}} \tag{78}$$

$$\frac{\partial^2 E}{\partial x_j^{(m+1)2}} = \frac{\partial^2 E}{\partial o_j^{(m+1)2}} \left(\sigma'\left(x_j^{(m+1)}\right)\right)^2 + \frac{\partial E}{\partial o_j^{(m+1)}} \sigma''\left(x_j^{(m+1)}\right) \tag{79}$$

