# Peer review of "The Incredible Shrinking Neural Network: New Perspectives on Learning Representations Through The Lens of Pruning"

_ICLR 2017 — rejected_

[Author Response · Aditya Sharma · 03 Dec 2016]
**Revision History**

First Revision: 27 November, 2016. Added Section 4.5: Investigation of Pruning Performance with Imperfect Starting Conditions. We discuss the impact of pruning sub-optimally trained networks in this section by specifically analyzing performance of networks classifying the digits 0, 1 and 2 of the MNIST database.

[Author Response · Aditya Sharma · 10 Dec 2016 (modified: 15 Dec 2016)]
**Frequently Asked Questions**

Here are answers to some common questions the authors have been asked about the current work in the past by readers of the manuscript. We hope these will help clarify any other questions our reviewers/readers might have.

Q: Why doesn't the paper present numerical comparision to state-of-the-art/recent pruning techniques?

A: Under certain motivational assumptions, it is understandable to demand benchmarking comparisons against state-of-the-art methods, but this may be missing the fundamental purpose of the present research. Our investigation is intended less to propose a competing alternative to existing pruning techniques and more to shed light on the limitations of generally accepted approaches to pruning and the degree to which increased numbers of parameters affect learning representations in neural networks. The paper does talk about most, if not all popoular pruning techniques out there. In fact, we examined the literature for numerical methods to approximate the importance of network elements, and the widely-cited 1st & 2nd order techniques proposed by Mozer, LeCun, Hassibi, Stork, et al. provided our initial inspiration. This is the jumping off point for our research in terms of key insights.

Q: The idea of using Taylor series approximations seems interesting but not really effective.

A: It is not effective when used as a pruning technique but it is VERY effective to test out the effectiveness of existing pruning techniques, which is what we do here. We have mentioned it multiple times in the paper that the motivation behind this work is NOT to propose a new pruning technique that will outperform all other techniques out there but to tap into learning representations to see how effective our established techniques are when seen from the perspective of representations. The Taylor series approximations play an important role here. A lot of pruning techniques out there use 2nd Order error gradients and assume that using them is the most effective way to prune networks. We have conclusively proved using the Taylor series that this is very much not the case. Our results with the brute-force method show us that there is a much larger extent to which networks can be pruned. This makes for a great starting-off point for future research to find methods that can produce similar results.

Q: Why did you decide in favor of sigmoid activation functions instead of something more recent and more popular like ReLUs? 

A: As mentioned above, the main contribution of this work is to demonstrate the feasibility of pruning entire neurons from trained networks, and offer novel insight on learning representations. We use Taylor methods to approximate the results achieved by the brute-force method but this is not an ideal solution to the problem, as we discuss. The 2nd order approximation technique will not work for ReLU networks because ReLUs do not have a 2nd derivative, unless we use the soft-plus function as a continuous approximation. Furthermore, due to the fact that we are approximating the error surface of a network element with respect to the output using a parabola, if there is no useful parabola to approximate this relationship, then the method breaks down. The derivatives of the activation function are simply parameters of the Taylor series. It doesn’t cease to be a parabolic approximation or become more effective if we use a different doubly-differentiable activation function. 

Q: Why carry out your experiments on the MNIST dataset and not go for a larger and more practical image dataset?

A: All experiments were necessarily carried out on optimally trained networks (not counting Section 4.5, which specifically examines non-optimally trained networks), so there is no way to improve them. We derived the algorithm assuming the well-studied sigmoid activation function. Furthermore, the MNIST dataset is a de-facto standard for demonstrating the potential of new techniques. A different dataset, task, activation function, or network architecture will not change the trends we see in the results but could make the results less interpretable. 

Q: The best setting is Iterative Re-ranking with Brute Force removal which is too expensive.

A: The brute-force method is highly parallelizable, so time complexity is not necessarily a deal-breaker. Our focus is the proof of concept, and we intend to investigate potential speedups in future work. Also, since pruning is anyways a single step carried out after the training process is over (which usually takes orders of magnitude more time), this is potentially acceptable.

[Official Review · AnonReviewer3 · rating 3 · confidence 4 · 15 Dec 2016]
**Looking at feature representations from the point of pruning is an interesting topic, but the conclusions nor the focus of this paper are clear**

The paper introduces a new pruning method for neural networks based on the second-order Taylor expansion and compares the results against a first-order method and brute-force pruning. It performs experiments of the three methods on several toy examples - including a two-layer network on MNIST - and shows that the second-order method behaves much worse then the brute-force baseline. In addition, from the success of the brute-force pruning the authors conclude that the hypothesis of Mozer et al - that neurons either contribute to performance or cancel out the effect of other neurons - is probably correct.

The authors put in considerable effort to explain all details of the paper clearly and at length, so the content of the paper is accessible even to people novel to pruning methods. Additionally, the authors have very carefully answered all questions that were coming up through the pre-review and have been very responsive.

My major criticism is that the paper lacks focus, does not have a concrete conclusion and does not explain what it adds to the literature. To make this apparent, I here summarise each paragraph of the conclusion section:

Paragraph 1: We do not benchmark / Pruning methods do not fare well against brute-force baseline / Some evidence for hypothesis of Mozer & Smolensky, but further investigation needed

Paragraph 2: Introduced 2nd order Taylor method / Does not fare well against baseline

Paragraph 3: Re-training may help but is not fair

Paragraph 4: Brute-force can prune 40-70% in shallow networks

Paragraph 5: Brute-force less effective in deep networks

Paragraph 6: Not all neurons contribute equally to performance of network

The title of the paper and answers of the authors to the pre-review questions seemed to strongly suggest that the paper is not about the new second-order method, is not about benchmarking pruning algorithms but is instead about the learnt representations. But only two or three sentences in the conclusion, and no sentence in the part on results in the abstract, even refers to neural representations. In an answer to the pre-review questions the authors stated:

> Furthermore, we do not have to accept the conclusion that re-training is a necessary part of pruning because a brute force search reveals that neurons can in fact be 
> pruned from trained networks in a piecemeal fashion with no retraining and minimal adverse effect on the overall performance of the network. This would be 
> impossible if neurons did not belong to the distinct classes we describe."

But this can already be concluded from the 2nd order method, which has a similar characteristic and is based on other 2nd order methods (not shown here). What is the motivation to introduce a new 2nd order method here?

In addition, some other minor conclusions about representations - in particular the cancellation effect - might be based on side-effects of the greedy serial pruning method. Optimally, one would need to consider all the different ways of pruning (which, of course, scales exponentially with the number of neurons and is computationally infeasible). Notably, the authors do consider this limitation in the context of conventional pruning methods in the conclusions: "Third, we assumed that pruning could be done in a serial fashion [...]. We found that all of these assumptions are deeply flawed in the sense that the true relevance of a neuron can only be partially approximated [...] at certain stages of the pruning process". But the brute-force pruning process is also serial - why is that not a problem?

All in all it is unclear to me what the paper adds: there are little conclusions regarding the learnt representations nor is there sufficient benchmarking against state-of-the-art pruning methods. I would suggest to focus the paper in the following way: first, use a state-of-the-art pruning method from the literature (that works without re-training) or do not use any other pruning methods besides brute-force (depending on whether you want to compare pruning methods against brute-force, or want to learn something about the learnt representations). In this way you need to write little about this second-order tuning methods, and readers are not so easily confused about the purpose of this paper (plus it will be considerably shorter!). Then concentrate on 2-layer MNIST and a deeper CIFAR10 network. Further focus the paper by adding an itemised list of the exact contributions that you make, and streamline the paper accordingly. These measures could strongly boost the impact of your work but will require a major revision.

PS: I think the confusion starts with the following sentence in the abstract: "In this work we set out to test several long-held hypothesis about neural network learning representations and numerical approaches to pruning." Both aspects are pretty orthogonal, but are completely mixed up in the paper.

[Official Review · AnonReviewer2 · rating 3 · confidence 4 · 16 Dec 2016]
**No Title**

I did enjoy reading some of the introductions and background, in particular that of reminding readers of popular papers from the late 1980s and early 1990s. The idea of the proposal is straight forward: remove neurons based on the estimated change in the loss function from the packpropagation estimate with either first or second order backpropagation. The results are as expected that the first order method is worse then the second order method which in turn is worse than the brute force method.

However, there are many reasons why I think that this work is not appropriate for ICLR. For one, there is now a much stronger comprehension of weight decay algorithms and their relation to Bayesian priors which has not been mentioned at all. I would think that any work in this regime would require at least some comments about this. Furthermore, there are many statements in the text that are not necessarily true, in particular in light of deep networks with modern regularization methods. For example, the authors state that the most accurate method is what they call brute-force. However, this assumes that the effects of each neurons are independent which might not be the case. So the serial order of removal is not necessarily the best. 

I also still think that this paper is unnecessarily long and the idea and the results could have been delivered in a much compressed way. I also don’t think just writing a Q&A section is not enough, and the points should be included in the paper.

[Official Review · AnonReviewer1 · rating 3 · confidence 4 · 17 Dec 2016]
**a good effort at understanding representations via pruning, but unclear overall contribution, with less than necessary results and methodological novelty**

The authors have put forward a sincere effort to investigate the "fundamental nature of learning representations in neural networks", a topic of great interest and importance to our field.  They propose to do this via a few simplistic pruning algorithms, to essentially monitor performance decay as a function of unit pruning.  This is an interesting idea and one that could potentially be instructive, though in total I don't think that has been achieved here.  

First, I find the introduction of pruning lengthy and not particularly novel or surprising.  For example, Fig 1 is not necessary, nor is most of the preamble section 3.3.0.  The pruning algorithms themselves are sensible (though overly simplistic) approaches, which of course would not matter if they were effective in addressing the question.  However, in looking for contributions this paper makes, an interesting, pithy, or novel take on pruning is not one of them, in my opinion.

Second, and most relevant to my overall rating, Section 4 does not get deeper than scratching the surface.  The figures do not offer much beyond the expected decay in performance as a percentage of neurons removed or gain value.  The experiments themselves are not particularly deep, covering a toy problem and MNIST, which does not convince me that I can draw lessons to the broader story of neural networks more generally.  

Third, there is no essential algorithmic, architectural, or mathematical insight, which I expect out of all but the most heavily experimental papers.

[Public Comment · (anonymous) · 02 Feb 2017]
**Some related works**

1) Wen, Wei, et al. "Learning structured sparsity in deep neural networks." Advances in Neural Information Processing Systems. 2016.
2) Lebedev, Vadim, and Victor Lempitsky. "Fast convnets using group-wise brain damage." Proceedings of the IEEE Conference on Computer Vision and Pattern Recognition. 2016.
3) Alvarez, Jose M., and Mathieu Salzmann. "Learning the Number of Neurons in Deep Networks." Advances in Neural Information Processing Systems. 2016.

[Final Decision · Program Chairs · 06 Feb 2017]
**ICLR committee final decision**

The paper does not seem to have enough novelty, and the contribution is not clear enough due to presentation issues.